



# Moho and uppermost mantle structure in the greater Alpine area from S-to-P converted waves

Rainer Kind[1,2], Stefan M. Schmid[3], Xiaohui Yuan[2], Benjamin Heit[2], Thomas Meier[4] and the AlpArray and AlpArray-SWATH-D Working Groups[5]

[1] Freie Universität Berlin, Berlin
   [2] Deutsches GeoForschungsZentrum GFZ, 14473 Potsdam
   [3] Eidgenössische Technische Hochschule Zürich
   [4] Christian-Albrechts-Universität zu Kiel
   [5] A full list of authors appears at the end of the paper

*Correspondence to*: Rainer Kind (kind@gfz-potsdam.de)

**Abstract.** In the frame of the AlpArray project we analyze teleseismic data from permanent and temporary stations of the greater Alpine region to study seismic discontinuities down to about 140 km depth. We average broadband teleseismic S waveform data to retrieve S-to-P converted signals from below the seismic stations. In order to avoid processing artefacts, no

deconvolution or filtering is applied and S arrival times are used as reference. We show a number of north-south and east-west profiles through the greater Alpine area. The Moho signals are always seen very clearly, and also negative velocity gradients below the Moho are visible in a number of profiles. A Moho depression is visible along larger parts of the Alpine chain. It reaches its largest depth of 60 km beneath the Tauern Window. The Moho depression ends however abruptly near about 13°E below the eastern Tauern Window. The Moho depression may represent the mantle trench, where the Eurasian

lithosphere is subducted below the Adriatic lithosphere. East of 13°E an important along-strike change occurs; the image of the Moho changes completely. No Moho deepening is found in this easterly region; instead the Moho is updoming along the contact between the European and the Adriatic lithosphere all the way into the Pannonian Basin. An important along strike change was also detected in the upper mantle structure at about 14°E. There, the lateral disappearance of a zone of negative P-wave velocity gradient indicates that the S-dipping European slab laterally terminates east of the Tauern Window in the

axial zone of the Alps. The area east of about 13°E is known to have been affected by severe late-stage modifications of the structure of crust and uppermost mantle during the Miocene when the ALCAPA (Alpine, Carpathian, Pannonian) block was subject to E-directed lateral extrusion.

## 1 Introduction

The complicated tectonic structure of the Alpine mountain belt is a result of the collision of the African and European plates.

A summary of tectonics, seismic investigations and still fundamentally open questions is given by a description of the recent AlpArray project by Hetényi et al. (2018a). One of the main goals of the AlpArray project is revealing the deep structure of



the Eastern Alps. A comprehensive Moho map of the Alpine area has been published by Spada et al. (2013) combining controlled source seismics, receiver functions and local earthquake tomography. Earlier crustal models from controlled source studies have been published for example by Yan and Mechie (1989), Behm et al. (2007) and Grad et al. (2009a and

b).  Receiver functions have been used e.g. by Lombardi et al. (2008), Bianchi et al. (2015) and Brückl et al. (2010), who published integrated models of the crust in the Eastern Alps. Most controlled source and receiver function data from the Eastern Alps indicate, like in the Central Alps, southward subduction of the European plate. In contrast, mantle tomography results in the Eastern Alps either indicates a change of subduction polarity by revealing a north-dipping slab or, alternatively, cannot resolve the polarity clearly (Lippitsch et al., 2003, Molinari et al., 2015; Guidarelli et al., 2017; Kästle et al., 2018,

2020). Also, the presence of both, an Eurasian southward dipping slab and an Adriatic northward dipping slab beneath the Eastern Alps have been proposed by Babuška et al. (1990) and only down to a depth of about 150 km by Kästle et al. (2020) and El-Sharkawy et al. (2020). A change of the subduction direction in the Eastern Alps is supported by geological arguments (Schmid et al., 2004; Kissling et al., 2006; Handy et al., 2015). In the last years large-scale international seismic experiments were carried out (AlpArray: see Hetényi et al., 2018a and SWATH-D: see Heit et al., 2017, 2021) and provided

large amounts of new data. We used these new data from broadband stations together with data from earlier temporary experiments and from all available permanent broadband stations. We also processed the seismic record using a novel method. This new method uses S-to-P conversions, similar to the S-receiver function method, except that raw untouched broadband data are stacked without any filtering or deconvolution. This method helps to avoid processing artefacts related to filtering (Kind et al., 2020). The S arrival times are automatically picked and used as reference.

## 2. Data and Method

All data used have been copied from the EIDA portals (e.g. http://www.orfeus-eu.org/data/eida). We selected events with magnitudes greater than 5.5 within the epicentral distances between 60° and 85°. Broadband data in time windows of 400 seconds before and after the theoretical S onset have been copied from the data archives. The distribution of stations used in the greater Alpine area is shown in Fig. 1. All networks contributing data are listed in the Acknowledgments. The basic idea

was to stack nearly untouched common conversion point signals in the time domain with the arrival time of the SV signal as reference. A number of processing steps are required. The three-component traces for each record are rotated from the Z, N, E coordinate system using the theoretical backazimuth and incidence angles according to the IASP91 model (Kennett and Engdahl, 1991) into the local ray coordinate system P, SV, SH (also called the L,Q,T system). We automatically picked the first arrivals of the SV signal (Baer & Kradolfer, 1987) using data with a signal-to-noise ratio greater than 6 and corrected

for the sign of the onset. The magnitude of the events was only considered for copying data from the archives. We normalized the traces with the absolute maximum within a window of 10 s after the S onset on the SV component. To ensure high-quality data we chose only events for which the amplitudes on the P component are less than 50 per cent of the input SV signal on the SV component within the time window of −50 to −10 s before the SV onset. The reason for applying this





criterion is that very large signals on the P component within that window can only be regarded as noise. At the same time,

by lowering this limit too much we have significantly less data and may not be able to obtain a sufficient number of traces for summation. This is because the converted signals of interest are in most cases below the noise level in the individual traces. The expected S-to-P converted Moho signals are close to 10% of the incident signal and just a few percent in conversions from below the Moho. After applying these data quality criteria, more than 400,000 three-component records remained for analysis. To reduce the amount of data we resampled all traces to 10 samples per second by linear interpolation.

Before summation in time domain a distance moveout correction is applied in which we expanded or stretched the time scale to simulate a slowness of 6.4 s/° of the incoming wave. This is the slowness value frequently used in P and S receiver function studies. S-to-P piercing point coordinates at depths of 50 and 100 km have been computed for the common piercing point (CCP) summation. The IASP91 model was used for these two processing steps. To process the data, we used the Seismic Handler software package (Stammler, 1993; www.seismic-handler.org).

**3. The Profiles**

In order to obtain high resolution along the profiles we chose a size of the CCP area of 0.2° in a direction parallel to the profiles. However, to keep the CCP area large enough for a sufficient number of traces used for the summation, we increased the extension of the CCP area perpendicular to the profile up to 1.0° or more. Profiles with a smaller extension of the CCP area perpendicular to the profile are shown in the Supplementary Material for comparison. The locations of the chosen

profiles are displayed in Fig. 2 and the corresponding seismic sections are shown in Figs. 3-14. The panels on the left side of the figures (labeled A) are for piercing points at 50 km and the panels on the right side (labeled B) are for piercing points at 100 km depth. This means that the summation traces are best focused at these particular depths. The traces in the same geographical bins between A and B are not identical. Consequently, the left-hand panels focus on the Moho depth while the right-hand panels point at discontinuities below the Moho. Each trace results from summing of at least 200 traces. The

precursor time is given on the left scale relative to the SV arrival time. A depth scale is given on the right side of each figure. These depths were computed using the one-dimensional IASP91 global reference model. We provided the time scale in order to enable the use of alternative models for depth calculation. The amplitude scale is given relative to the amplitude of the incident SV signal. The latitude or longitude of the center of each bin is given at the top of each figure. There is no overlapping between neighboring traces along the profiles. The width of the profiles is given at the bottom of each figure.


The signals converted from S to P at the Moho are blue and denote an increase of velocity with increasing depth. Red signals denote a downward velocity decrease. The arrival times of the seismic signals must be picked as usual in seismograms at the beginning of the signals and not at the signal maximum, as is usually the case for the classical receiver function method. We marked approximate Moho arrival times by dotted lines. The signal forms of the Moho (and other) conversions are

determined mainly by the signal forms of the incident SV signals. The response of the structure below the station (mainly the





Moho interface) is also contributing to the signal form. Below the Moho in the right-hand panels we see, for most of the profiles, concentrations of red signals. These are caused by sharp or gradual downward velocity decreases. These signals below the Moho appear scattered, and lack clear lateral correlations, when compared to the Moho signals, that align much better. For this reason, we refer to the red regions below the Moho as zones of Negative Velocity Gradients (NVG). In some 100 profiles we marked the bottom of the NVGs by a broad grey line that approximately indicates the earliest arrival times from the deepest scatterers. A negative seismic velocity gradient is often expected to mark the lithosphere asthenosphere boundary. But its location depends on the nature of the geophysical measurements used to locate this boundary. Apart from temperature effects this negative gradient may also be caused by changes in, for example, chemical composition or anisotropy (e.g. Eaton et al., 2009). Therefore, we prefer to use the neutral term NVG here.

## 4. Characteristic Observations Along the Profiles

### 4.1. Moho Structure

#### 4.1.1. North-south profiles

We first present a series of north-south oriented profiles shown in Figs. 3-11. In the entire region between 8.0 to 13.5°E (profiles 2-6 in Figs. 4A-8A) we see a south dipping European Moho marked by black dotted lines. The Moho reaches its 110 largest depth in all profiles 2-6 at about 46.5° N. This is near the latitude at which Spada et al. (2013) draw the boundary between European and Adriatic Moho, respectively (see black line in Fig. 2) that roughly coincides with the location of the E-W striking part of the Periadriatic line west of longitude 13°E. This implies that, according to geological evidence (Schmid et al. 2004; Kissling et al. 2006), the Moho south of about 46.5° N, depicted in the profiles 2-6, is the Adriatic Moho. In profiles 2-6 this Adriatic Moho rises towards a culmination located near 45.5°N before descending southward beneath the 115 Ligurian Moho at the front of the Apennines. This is shown by the onsets of the Adriatic Moho, which is also marked by dotted black lines (Figs. 4A-8A). At first sight, this geometry suggests a wedge-shaped trough below the Alpine chain. In fact, the two Moho's are separated from each other by a subduction corridor, whereby the European Moho dips undisputedly southward beneath the Adriatic Moho according to geological and geophysical evidence (Schmid et al. 2004; Kissling et al. 2006), at least west of 11°E. A far-reaching European Moho below the Adriatic Moho is, however, not observed in our data. 120 This is different to the continental collision observed between India and Asia in Tibet where a double Moho is observed and is interpreted as Indian crust reaching several hundred kilometers below the Tibetan crust (Yuan et al. 1997, Nabelek et al. 2009). In the Alps, east of 11°E opinions about the subduction polarity have remained controversial so far (see discussion in Kästle et al. 2020). The maximum depth of the Moho reaches about 50 km near 8.5°E (Fig. 4A) and increases to about 60 km near 13°E (Fig. 8A). A very similar maximum Moho depth is also observed in the west near 7.5°E. Here the Moho depth 125 reaches about 65 km at about 45.5°N (see profile 1 Fig. 3A). This Moho interface is hence located further south, which is caused by the south bending of the Alpine chain in the west (Fig. 2).



A significant change in the structure of the Moho below the Alps is observed near 13° E, i.e. near the eastern margin of the Tauern Window by comparing the Fig. 8A with Fig. 9A. West of this longitude the maximum depth of the Moho depression is around 60 km at 47°N (black arrow in profile 6 of Fig. 8A). East of 13.5°E the Moho depth at 47°N is only about 40 km (black arrow in profile 7 of Fig. 9A), and the Moho depression near latitude 47° seen in Fig. 8A is suddenly replaced by a culmination at the same latitude. This implies that the Moho topography no longer reflects a depression that would be expected to coincide with a scenario of subduction of the European plate below the Adriatic plate (or the opposite) in the Alps east of about 13°E.

On the other hand, a so far unknown substantial Moho depression similar to that observed at the plate interface west of 13°E is observed further in the north (at 48-49°N) in the profiles east of about 13°E (see profiles 7-9 in Figs. 9A-11A). In the case of profiles 7 and 8 this Moho depression is located well north of the Alpine front (marked with red arrows in Figs. 9A and 10A; see also Fig. 2) and beneath the southern Bohemian Massif belonging to the European plate. In profile 9 the deepest point is below the external Western Carpathians. In this very wide (between 17-19°E) profile 9 a big jump in the Moho onset time is seen near 47°N marked with a black arrow in Fig. 11A. This occurs at the location of the Mid-Hungarian Fault Zone (MHZ in Fig. 2, e.g. Hetényi et al. 2015). North of this zone the Moho is at about 30 km and deepens to 50 km underneath the frontal West Carpathians. South of the MHZ the Moho suddenly rises to only some 20 km depth beneath the Pannonian Basin south of Lake Balaton. The influence of potentially different velocity models is not considered in the Moho depth estimates discussed above. Fig. 15 summarizes locations and maximum depths of the Moho along the Alpine chain in map view.

### 4.1.2. East-west profiles

In a second step we discuss the Moho depth shown along three east-west profiles presented in Figs. 12A-14A. Fig. 12A shows profile 10 centered at 47°N and entirely located on the European plate east of longitude 11.5°E but straddling along the plate boundary with Adria further to the east. This profile reaches the maximum Moho depth of around 60 km at about 13°E, i.e. in the area of the Tauern Window (see Fig. 2). Eastward, the Moho depth of this profile steadily rises to less than 20 km beneath the Pannonian Basin. West of its deepest point beneath the Tauern Window the Moho also rises, but exhibits a secondary depression between 7-8°E located in the European plate near the Alpine Front (Fig. 2). For easier comparison the dotted black line marking the Moho trend in Fig. 12A is copied into the neighboring profiles 11 and 12 located north and south of profile 10, respectively (Figs. 13A and 14A). Profile 11 (Fig. 13A) located further north at 48°N, entirely runs within the European plate and is similar to the one at 47°N (Fig. 12A). However, in its western part the Moho is generally shallower in profile 11, probably due to the proximity to the Rhine Graben. Conversely, in the eastern part of this profile 11 the Moho is generally deeper than that of profile 10 at 47°N. This confirms the existence of a substantial Moho depression beneath the Bohemian Massif, which was found in the N-S profiles 7 and 8 (Figs. 9A and 10A). This profile 11 (Fig. 13A) also shows that the shallowing of the Moho towards the east is by far more moderate compared to that seen in profile 10. The





southernmost E-W profile 12 at 46°N (Fig. 14A) runs within the Adriatic plate east of longitude 8.5° and shows a Moho
trend that is very similar to the one at 47°N (Fig. 12A). Only in its western part, running within the European part, the Moho
is somewhat deeper.

## 4.2. Structure within the uppermost Mantle

The left and right panels of Figs. 3-14 show profiles of summed traces, which have their piercing points within the same
geographical bins. However, the piercing points are computed for different depths, namely 50 km for the left hand panels
and 100 km for the right hand panels. Therefore, the summed traces in the bins are not identical. We now focus on the right
panels computed by choosing a piercing point of 100 km, which is optimal when searching for structures in the shallow
mantle below the Moho. Almost all Figs. 3B-14B show a relatively large number of negative signals (marked in red)
indicating downward velocity reductions at some depth below the Moho. However, there are significant differences between
these red signals compared to the blue signals marking the Moho. The amplitude of the Moho signals is nearly 10% of the
incident SV wave, whereas the amplitude of the negative signals from below the Moho is in most cases below 4%.
Moreover, in most cases the Moho signals mark a clear discontinuity, which can be correlated over the entire length of the
profiles, whereas the negative signals from below the Moho do not mark a clear discontinuity but appear much more
scattered. However, in parts of some of the profiles we were able mark the lower boundary of regions with exceptionally
high concentrations of negative signals with a scattered grey line. We refer to such red regions as Negative Velocity Gradient
zones (NVG). Similar to the Moho, the bottom of such NVG zones marked by the broad grey line indicates the arrival times
of the deepest scatterers.

### 4.2.1. North-south profiles

An NVG could be mapped in profile 1 (Fig. 3B), south dipping down to about 110 km. In profiles 2-4  (Fig. 4B-6B, between
8-11°E) the NVG is horizontal or slightly inclined to the south. The base of the NVGs is found at depths of 80-90 km, and
they end towards the south at around latitude 47° to 48°, i.e. at or near the front of the Alps (see Fig. 2). This shows that they
are  definitely embedded in the European plate of the Alpine foreland dipping southwards underneath the front of the Alps.
In profiles 5-7 (Figs. 7B-9B, between 11-14.5°E), the NVGs appear more pronounced. They are again south dipping and
their base reaches down to 115-130 km depth. They also end southward at around latitude 47°, which is somewhat further
inside the Alps given the SSW-ENE strike of the Alpine front but still within the European plate (see Fig. 1). However, there
is practically no significant indication of an NVG in profile 8 (Fig. 10B, 14.5-17°E), i.e. in the area located well east of the
Tauern Window spanning the Alps-Carpathians transition area (see Fig. 1). Note that profile 8, together with profile 9, was
chosen relatively wide because of the scarcity of stations in the area. In the easternmost N-S profile 9 east of 17°E (Fig. 11B)
running across the Western Carpathians and across the MHZ, a very different structure of the NVG is seen. The  NVG is
subhorizontal north of about 48.5°N, where the base of the NVG shows a significant step moving up from 90 km beneath the
European foreland and the external Carpathians to 60 km beneath the internal Carpathians and the Pannonian plain. We note





that this jump in the depth of the NVG closely coincides with the onset of massive thinning of the lithosphere in the wider Pannonian realm that also affects the Central Western Carpathians south of about 48.5° (see Fig. 10 of Horváth et al. 2006). This contrasts with the step of the Moho in Fig. 11A that is more moderate (about 10 km) and takes place further to the

south, namely across the MHZ at 47°N. This points to differences in amount and locus of thinning in the crust and mantle lithosphere, a fact that is well known in the Pannonian area where the thinning of the mantle lithosphere by far exceeds crustal thinning (e.g. Horváth et al. 2006).

### 4.2.2. East-west profiles

The east-west profile 10 centered along 47°N (Fig. 12B) shows a very clear NVG that steeply dips eastward and reaches

about 130 km depth at 13°E where it turns horizontal and disappears near 14E. Comparing this easterly dip of the NVG in an E-W profile with its southerly dip in the N-S profiles at 11°-14.5°E (Figs. 7-9) shows that the NVG is in fact dipping towards the southeast. The east-west profile 11 along 48°N (Fig. 13B) along the northern Alpine front also shows an apparently east dipping NVG, however only reaching about 110 km depth at around 12°E. Profile 12 along 46°N (Fig. 14B) below the Po valley and located within the Adria plate in its eastern part, however, lacks a similar east dipping NVG. It

seems astonishing that relatively steep dipping structures can be imaged by steeply incident converted waves. Our interpretation is that the NVG area may contain a sufficient number of horizontal scatterers, which can be observed. The dip angle of the NVG might be biased (Schneider et al., 2013). Fig. 16 summarizes the deepest locations of the bottom of the NVG in the Spada (2013) Moho map and shows that they are all located in the area of the European plate. These deepest points are located (1) in the northern Alpine foreland (Bohemian Massif), (2) in the frontal part of the West Carpathians and

(3) beneath the axial zone of the central and Eastern Alps, where they reach depths between 110 km and 130 in an area close to the southern margin of the European plate and in the vicinity of its contact with the Adria plate.

### 5. Comparison with Earlier Results

The comparisons of our results with those obtained by previous studies are discussed here and are displayed in Figs. 17 to 20.


In Fig. 17A we compare our Moho data obtained along the north south profile 6 centered at 13°E (Fig. 8) with the results of Hetényi et al. (2018b) and Bianchi et al. (2020) along their profile at 13.3°E (temporary passive experiment EASI). Hetényi et al. (2018b) used P receiver function data from this profile between the Eger Rift in the northern part of the Bohemian Massif and the Adriatic Sea. Their recorded Moho depth is plotted as cyan dotted line along our section 6 in Fig. 17A. Their

Moho depth increases from 25 km at the Eger Rift at the northern end to about 30 km at the Alpine front at 48°N. From there the European Moho deepens rapidly to about 60 km at 47°N below the Tauern Window. The northward dipping Adriatic Moho reaches deeper down to 70 km and dips below the European Moho according to these authors. This is not seen in our





results. The very general shape of the Moho is nevertheless in good agreement with our data. However, their depths of the European and Adriatic Moho are about 5 km shallower than ours, except near the deepest point at 47°N. This might partly be

due to the different velocity models used. Hetényi et al. (2018b) used a local model whereas we used a global model (IASP91). Our data do not support the postulate of Hetényi et al. (2018b) that the Adriatic Moho in the Eastern Alps dips northward underneath the European Moho. Instead, our Moho depression looks rather symmetric similar to Spada et al. (2013). Bianchi et al. (2020) used PKP multiples within the crust below the seismic stations to derive the Moho depth. Their results are marked by a slight grey dotted line in Fig. 17A. The agreement between our data with those of Bianchi et al.

(2020) is also reasonably good, except near 47°N, where they have no data. It should also be noted, that the "Moho gap" mapped in white by Spada et al. (2013) at 47°N (Fig. 2) does not seem to exist according to our data, and those of Mroczek et al. (2020) and Hetényi et al. (2018b).

In Fig. 17B we compare our data in the mantle below the Moho with results from the teleseismic body wave tomography by

Paffrath et al. (2021) by superimposing the P-wave velocity contours that these authors used for calculating velocity anomalies along our E-W profile 10. The E-dipping base of the red NVG zone roughly coincides with the top of an area associated with abnormally low P-wave velocities below 8.0 km sec-1 marked in yellow in the western part of this profile, but only until 12° longitude. At this point it has to be mentioned that this E-W profile 10 runs at an acute angle to the strike of the Alpine front running SW-NE to WSW-ENE between the arc of the Western Alps and the Tauern Window (see Fig. 2),

and hence it records a distorted profile across the SSE-dipping European plate. The same NVG zone was also traversed by the N-S profiles that showed the NVG zone to be hosted within the SSE-dipping European plate. The termination of the base of the east dipping NVG zone turning flat at 13° longitude (green line in Fig. 12B) and terminating at around 14°E hence strongly suggests that the European slab is not reaching the area covered by this E-W profile east of 14°E, i.e. near the eastern margin of the Tauern Window. East of 12°E at a depth of 110 km the P-wave velocity contours from Paffrath et al.

(2021) actually indicate a low velocity zone that rises to a shallower depth towards the east, together with the rise of the Moho seen at the base of the blue Moho amplitudes in this figure (and marked also in Fig.10A focusing on the Moho depth). This again points towards an important along strike change in the area of the Tauern Window in the axial zone of the Alpine orogen that will be discussed later. In the context of this comparison it should however be noted that our method of using converted waves is sensitive to velocity gradients and not absolute velocities. This means that smaller scale relative velocity

reductions could well be embedded in larger high or low velocity regions. Fig.17B demonstrates that our observations of the NVG reaching down to ca. 140 km depth is in agreement with the tomographic low velocity anomaly documented for the area west of longitude 11°E. Between 11°E and 14°E, a downward positive velocity gradient is found by Paffrath et al. (2021) where our data indicate a NVG (see map in Fig. 16 and profile 10 in Fig. 17B). Further studies and high-resolution S-wave velocity models are needed to resolve this discrepancy. The maximum recorded depths of the NVG anomaly given in

the map in Fig. 16 indicate that its southern end becomes deeper towards the east, confirming what can be directly seen by inspection of the E-W profile of Fig. 9B.





Fig. 18A compares our north-south Moho profile 6 with the Eastern Alps Moho profile by Brückl et al. (2010) at 13°E. The agreement of the profiles is very good. The general conclusion of Brückl et al. (2010) that the European Moho is subducting

below the Adriatic Moho is not obvious from our data, but the asymmetry in our data is in favor of this conclusion. Our data compare fairly well with the data of Kummerow et al. (2004) along their profile at 12°E although their Moho depths appear shallower. In Fig. 18B we compare our north-south profile along a corridor between 10 and 12°E with a profile in the map of Spada et al. (2013) (see Fig. 2) at 11°E. The agreement between the two data sets is very good. The Moho jump near 44°N marking the transition from the Adriatic to the Ligurian Moho is evident on both profiles.


In Fig. 19A we compare our Moho data along profile 9 (Fig. 11A) depicting a culmination of the Moho with the RF data of Hetényi et al. (2015) by projecting the southwest-northeast profile of Hetényi et al. (2015) onto our north-south profile. This same Moho culmination follows the strike of the MHZ and was traced further to the ENE by Kalmar et al. (2019). North of the MHZ the Moho is rapidly deepening towards the north. Fig. 19B compares our profile 9 (Fig. 11B) with mantle

tomographic results from Paffrath et al. (2021), geologically interpreted by Handy et al. (2020). The NVG in Fig. 19B to both sides of the MHZ, which is very shallow and located immediately below the Moho, is not changing depth across the MHZ at 47°N where the Moho starts to deepen (see Fig. 19A). Rather, an offset of the NVG to greater depth occurs at around 48.5N, which is located within the Central West Carpathians, i.e. north of the Pannonian Basin. The comparison with the tomographic data shows that this jump in depth of the NVG coincides with the location of the Alpine-Tethys suture zone

identified in tomography data (Handy et al. 2020), a zone that forms the southern boundary of the European lithosphere (green dotted subvertical line in Fig. 19B). Our data reinforce the fundamental difference between the seismic properties of the European lithosphere beneath the frontal Western Carpathians with those of the more internal parts of the Western Carpathians and the Pannonian realm characterized by a shallow lithosphere-asthenosphere boundary (e.g. Horváth et al. 2006; Kind et al., 2017).


In Fig. 20A we compare our Moho depth profile along the east west profile 10 at 47°N (Fig. 12A) with early results of seismic refraction experiments (Yan and Mechie, 1989). The agreement in the general trend of eastward shallowing starting in western Tauern Window at around 12°E is good, although there are some differences in details. Fig. 20B presents an additional profile 13 located north of profile 11 at 49-50°N in the European lithosphere crossing the front of the Western

Carpathians at 17.5°E (Fig. 2), added for comparison with the Moho map of Grad et al. (2009a), based on a variety of geophysical data. The black dotted line in this Fig. 20B marks the rapid eastward Moho deepening towards the front of the Western Carpathians initiating at ca. 14°E and reaching 50 km below the external Western Carpathians. The discrepancy with the data of Grad et al. (2009a) is considerable in the easternmost part of this profile where our data show a much more pronounced deepening of the Moho. This eastward deepening along the northerly profile of Fig. 20B strongly contrasts with

the eastward shallowing of the Moho shown in Fig. 20A, i.e. in a more southerly profile (at 46.5-47.5°N) crossing the area of



the northern margin of the Pannonian plain around the MHZ. This marked difference is due to the fact that in our data the Moho in the profile of Fig. 20B located in the European lithosphere bends down across the SW-NE striking front of the Western Carpathians (see Fig. 2) while the profile of Fig. 20A runs within the Pannonian realm characterized by the eastward shallowing Moho depth shown in E-W profiles 10, 11 and 12, caused by Miocene backarc extension in the

Pannonian Basin, along the MHZ and in the internal parts of the Western Carpathians region (Horváth et al., 2015).

## 6. Summary of Observational Results

The comparison with results from other studies using other methods shows that by using broadband S-to-P converted signals we are able to obtain clear images of the Moho topography along most of the Alpine chain and also reveal the presence of some features that were not detected before. Furthermore, this method allows highlighting an important along-strike change

regarding Moho topography that takes place at around 13°E, i.e. in the eastern Tauern Window. The Moho-depression beneath the Central Alps along the Europe-Adria lithosphere boundary along 47°N deepens from some 50 km at 8°E to 60 km at around 13°E. In the N-S profiles the Moho depression very abruptly makes place to an up-doming of the Moho along the axial zone of the Alpine orogen at around 13°E, persisting all the way into the Pannonian Basin, as the depth of the Moho generally decreases to < 30 km also N and S of 47°N. East of 17°E rapid Moho deepening from about 20 to 30 km is

observed from south to north across the Mid-Hungarian Fault Zone. Our data also indicate the existence of a substantial Moho depression east of 13°E in front of the Alpine frontal thrust extending eastward into the area of the frontal Western Carpathians whose extent remained so far undetected (see Fig. 15).

In the shallow mantle we also observe, extending from 10°E to14°E, along an E-W section along 47°N combined with N-S

sections, steeply SSE-dipping areas of downward velocity reductions referred to as NVGs that extend to at least 140 km depth at 14°E and disappear east of there. The base of the NVG, which roughly coincides with the top of a low velocity zone revealed by mantle tomography west of 12°E (Paffrath et al. 2021) is related to the European slab. The termination of the NVG at 14°E suggests an important along strike change also in terms of the shallow mantle structure taking place at 14°E, i.e. about 1° east of the along strike change observed for the Moho-topography. Below the frontal West Carpathians at about

49°N we observed a rapid jump of the base of the subhorizontal NVG from about 60 km in the south to 90 km in the north.

## 7. Discussion and Conclusions

From a technical point of view the improved images of the present paper are, besides the obvious reason of using much larger amounts of data, also due to the modified data processing technique. In the receiver function technique seismic traces are sometimes filtered without paying sufficient attention to sidelobes or acausality. Another problem is the waveform

compression caused by deconvolution and summation along the maximum of the compressed signals, which also produces



sidelobes that tend to be misinterpreted. We therefore avoided all filtering and lined up the traces along the onsets of the S signals for summation. We found good agreements of our results with earlier results obtained in many regions. This emphasizes the reliability of our results. We also think that using additional seismic phases could help to increase the uniqueness and resolution of the seismic images and in consequence of the tectonic interpretation. A good example is the

usage of PKP multiples below the stations as shown by Bianchi et al. (2020).

From the geological point of view the Moho topography reflects deeper subduction of the European lithosphere only west of and in the Tauern Window area. This is no more the case east of 13° E, i.e. east of the eastern Tauern Window, where the Moho is up domed in the contact area between the European and the Adriatic lithosphere. According to geological evidence

(Ratschbacher et al. 1991) and according to the geological interpretation of recent mantle tomographic data (Paffrath et al. 2021, Handy et al., 2021) this updoming is likely due to a late stage modification of the Moho topography in the Eastern Alps, the Western Carpathians and the Pannonian Basin that occurred during the last ca. 20 Ma. Such updoming goes together with crustal thinning associated with lateral extrusion of the so-called ALCAPA mega-unit associated with the N-directed indentation of the Dolomites indenter and roll back of the Eastern Carpathians (Ratschbacher et al. 1991). The rapid

jump of the base of the NVG from about 60 km in the south to 90 km in the north below the frontal West Carpathians at about 49°N is also related to extension and mantle upwelling in the Pannonian Basin that also affected the internal Western Carpathians (Horváth et al. (2015).

Three main additional salient new features are of importance: (1) The abrupt offset of the Moho by about 10 km across the

Mid-Hungarian Fault Zone emphasizes the important role this fault zone played in terms of lateral and vertical offsets that occurred during the lateral extrusion of ALCAPA during the Miocene (Schmid et al. 2008), and, (2) A second Moho trough was detected north of the Alps, reaching down to about 50 km depth and located in the Bohemian Massif north of the Alpine front in the Eastern Alps, extending eastward into the frontal part of the Western Carpathians. This structure, hosted in the European lithosphere, has not been observed before and cannot be interpreted yet at this stage. Further studies are needed to

clarify this structure. (3) We observed areas of downward velocity reductions referred to as NVGs hosted within the European lithosphere in the foreland of the Alps at a depth of 80-90 km. In the Eastern Alps west of 14°E these NVGs can be traced down to depths of 115-140 km all the way to latitude 47°N, i.e. into the axial zone of the Alps.

**Team list.** Members of the AlpArray and AlpArray-SWATH-D Working Groups: György HETÉNYI, Rafael ABREU, Ivo

ALLEGRETTI, Maria-Theresia APOLONER, Coralie AUBERT, Simon BESANÇON, Maxime BÈS DE BERC, Götz BOKELMANN, Didier BRUNEL, Marco CAPELLO, Martina ČARMAN, Adriano CAVALIERE, Jérôme CHÈZE, Claudio CHIARABBA, John CLINTON, Glenn COUGOULAT, Wayne C. CRAWFORD, Luigia CRISTIANO, Tibor CZIFRA, Ezio D'ALEMA, Stefania DANESI, Romuald DANIEL, Anke DANNOWSKI, Iva DASOVIĆ, Anne DESCHAMPS, Jean-Xavier DESSA, Cécile DOUBRE, Sven EGDORF, ETHZ-SED Electronics Lab, Tomislav FIKET, Kasper FISCHER,





Wolfgang FRIEDERICH, Florian FUCHS, Sigward FUNKE, Domenico GIARDINI, Aladino GOVONI, Zoltán GRÁCZER, Gidera GRÖSCHL, Stefan HEIMERS, Ben HEIT, Davorka HERAK, Marijan HERAK, Johann HUBER, Dejan JARIĆ, Petr JEDLIČKA, Yan JIA, Hélène JUND, Edi KISSLING, Stefan KLINGEN, Bernhard KLOTZ, Petr KOLÍNSKÝ, Heidrun KOPP, Michael KORN, Josef KOTEK, Lothar KÜHNE, Krešo KUK, Dietrich LANGE, Jürgen LOOS, Sara LOVATI, Deny MALENGROS, Lucia MARGHERITI, Christophe MARON, Xavier MARTIN, Marco MASSA, Francesco

MAZZARINI, Thomas MEIER, Laurent MÉTRAL, Irene MOLINARI, Milena MORETTI, Anna NARDI, Jurij PAHOR, Anne PAUL, Catherine PÉQUEGNAT, Daniel PETERSEN, Damiano PESARESI, Davide PICCININI, Claudia PIROMALLO, Thomas PLENEFISCH, Jaroslava PLOMEROVÁ, Silvia PONDRELLI, Snježan PREVOLNIK, Roman RACINE, Marc RÉGNIER, Miriam REISS, Joachim RITTER, Georg RÜMPKER, Simone SALIMBENI, Marco SANTULIN, Werner SCHERER, Sven SCHIPPKUS, Detlef SCHULTE-KORTNACK, Vesna ŠIPKA, Stefano

SOLARINO, Daniele SPALLAROSSA, Kathrin SPIEKER, Josip STIPČEVIĆ, Angelo STROLLO, Bálint SÜLE, Gyöngyvér SZANYI, Eszter SZŰCS, Christine THOMAS, Martin THORWART, Frederik TILMANN, Stefan UEDING, Massimiliano VALLOCCHIA, Luděk VECSEY, René VOIGT, Joachim WASSERMANN, Zoltán WÉBER, Christian WEIDLE, Viktor WESZTERGOM, Gauthier WEYLAND, Stefan WIEMER, Felix WOLF, David WOLYNIEC, Thomas ZIEKE, Mladen ŽIVČIĆ, Helena ŽLEBČÍKOVÁ


**Author contributions.** Rainer Kind developed the initial idea of the project. Rainer Kind and Xiaohui Yuan did the data processing. Stefan Schmid and Thomas Meier did most of the interpretation and Ben Heit installed the stations in cooperation with the SWATH-D working group and participated in the interpretation

**Competing interests.** The authors declare that they have no conflict of interest.

**Acknowledgments**

This research was carried out within the AlpArray and SWATH-D projects which have been funded by the Deutsche Forschungsgemeinschaft within the Priority Program "Mountain Building Processes in Four Dimensions (MB-4D)" and by the GeoForschungsZentrum Potsdam. We are deeply indebted to all people involved in the instrument preparation, fieldwork

and data archiving. We thank numerous landowners for their willingness to host the stations, and all communities, authorities and institutes in the region for their great support of the project. Data of the SWATH-D network are archived and available from the GEOFON data center (Heit et al., 2017, 2021). AlpArray backbone data are archived at EIDA (http://www.orfeus-eu.org/data/eida). Instruments for the SWATH-D network were provided by the Geophysical Instrument Pool Potsdam GIPP of the GFZ Potsdam. Discussions with many colleagues within the Priority Program are greatly acknowledged. The tectonic

map was compiled by M.R. Handy. We also wish to thank Emanuel Kästle for his support. Most of the plotting has been done with GMT.





**The following networks have contributed data to this study:**

PASSEQ (7E), Albanian Seismological Network (AC), Belgian Seismic Network (BE), Bayernnetz (BW), Switzerland Seismological Network (CH), Croatian Seismograph Network (CR), Czech Regional Seismic Network (CZ), Danish
Seismological Network (DK), Estonian Seismic Network (EE), Northern Finland Seismological Network (FN), RESIF and other broadband and accelerometric permanent networks in metropolitan France (FR), GEOFON (GE), German Regional Seismic Network (GR), Regional Seismic Network of North Western Italy (GU), Finish National Seismic Network (HE), Hungarian National Seismological Network (HU), Global Seismograph Network - IRIS/USGS (IU), Italian National Seismic Network (IV), Mediterranean Very Broadband Seismographic Network (MN), Netherlands Seismic and Acoustic Network
(NL), Austrian Seismic Network (OE), North-East Italy Seismic Network (OX), Polish Seismological Network (PL), CEA/DASE Seismic Network (RD), Romanian Seismic Network (RO), Province Südtirol (SI), Serbian Seismological Network (SJ), Slovak National Seismic Network (SK), Seismic Network of the Republic of Slovenia (SL), Trentino Seismic Network (ST), SXNET Saxon Seismic Network (SX), Thüringer Seismologisches Netz (TH), Eifel Plume (XE), EASI Eastern Alpine Seismic Investigations (XT), Alparray (Z3), TOR-TE (ZA), SVEKALAPKO (ZB), TOR-TO (ZC), Transalp
II (ZO), SWATH-D (ZS), Bohema (ZV), JULS (ZW).

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





**Figure 1:** Distribution of permanent and temporary broadband stations used in this study (red inverted triangles) on a geological map of the area (after Hetényi et al. 2018a, their Fig. 1).




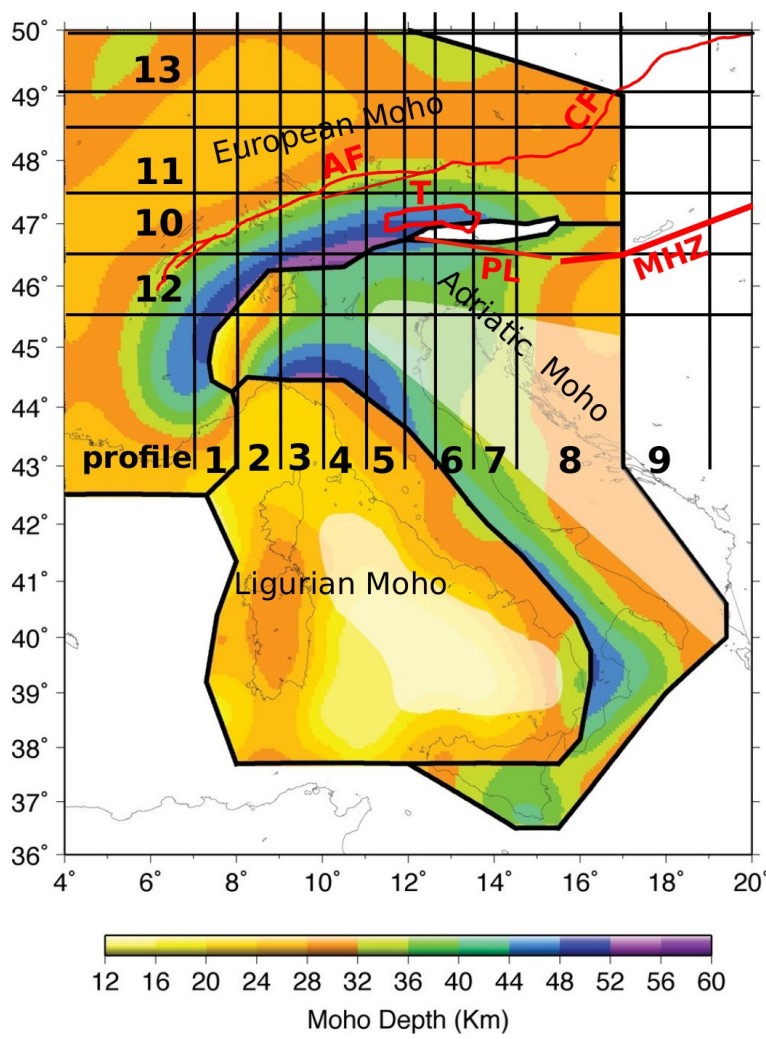

**Figure 2:** Moho map by Spada et al. (2013) with the locations of the sampling corridors used for profiles 1-13 shown in Fig. 3-14, 20. The locations of the Mid-Hungarian Fault Zone (MHZ) the Periadriatic Line (PL), the Tauern Window (T), the Alpine Front (AP) and the Carpathians Front (CP) are also marked. The space between two black lines marks the width of the profile corridors.




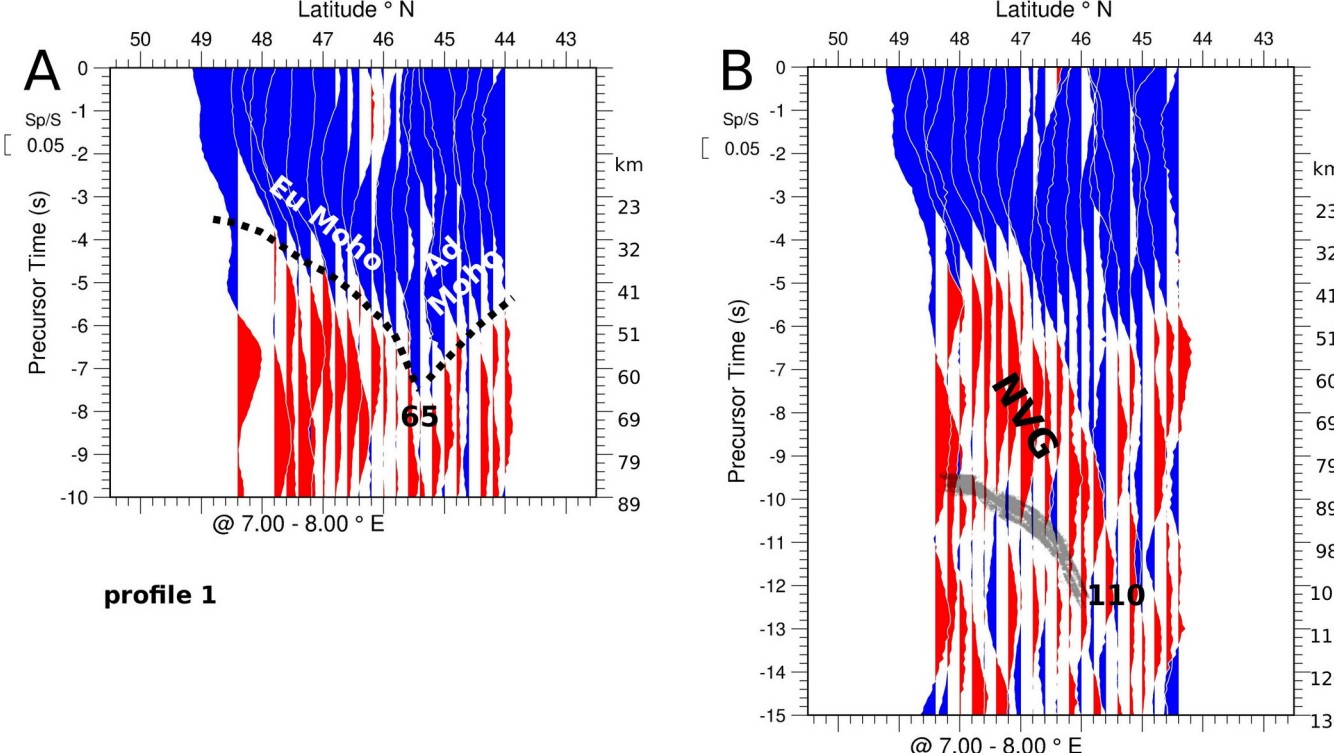

**Figure 3:** Profile 1: North-south profiles between 7-8°E. Differential times of converted waves relative to the S arrival times are given on the left of sections A and B. These times are converted into depth displayed on the right side of the sections using the global IASP91 velocity model. Blue colors indicate velocity increases with increasing depth (positive amplitudes) and red colors velocity decreases (negative amplitudes), respectively. Note vertical exaggeration of around 8.8 times. A) Section of summation traces within bins of S-to-P piercing points at 50 km depth, appropriate for imaging the Moho. Bin size along the profile is 0.2°, the bin size perpendicular to the profile is given at the bottom of the section. The onsets of the S-to-P converted signals at the Moho are approximated by a black dotted line. "Eu Moho" refers to the European Moho and "Ad Moho" designates the Adriatic Moho, respectively. The number given along the black dotted line in A indicates the greatest depth of the trough formed by the Moho signals. B) Same as in A, but configured for a piercing point depth of 100 km, appropriate for imaging the mantle lithosphere below the Moho. A concentration of negative (red) signals marked NVG (Negative Velocity Gradient) is visible below the Moho. The base of the NVG appears more scattered than the Moho. Therefore, the arrival times are only approximately marked by a scattered grey line. Similarly the number given at the bottom of the NVG in B approximately marks the deepest extend of the NVG. Legends of the following figures with profiles refer to this figure.





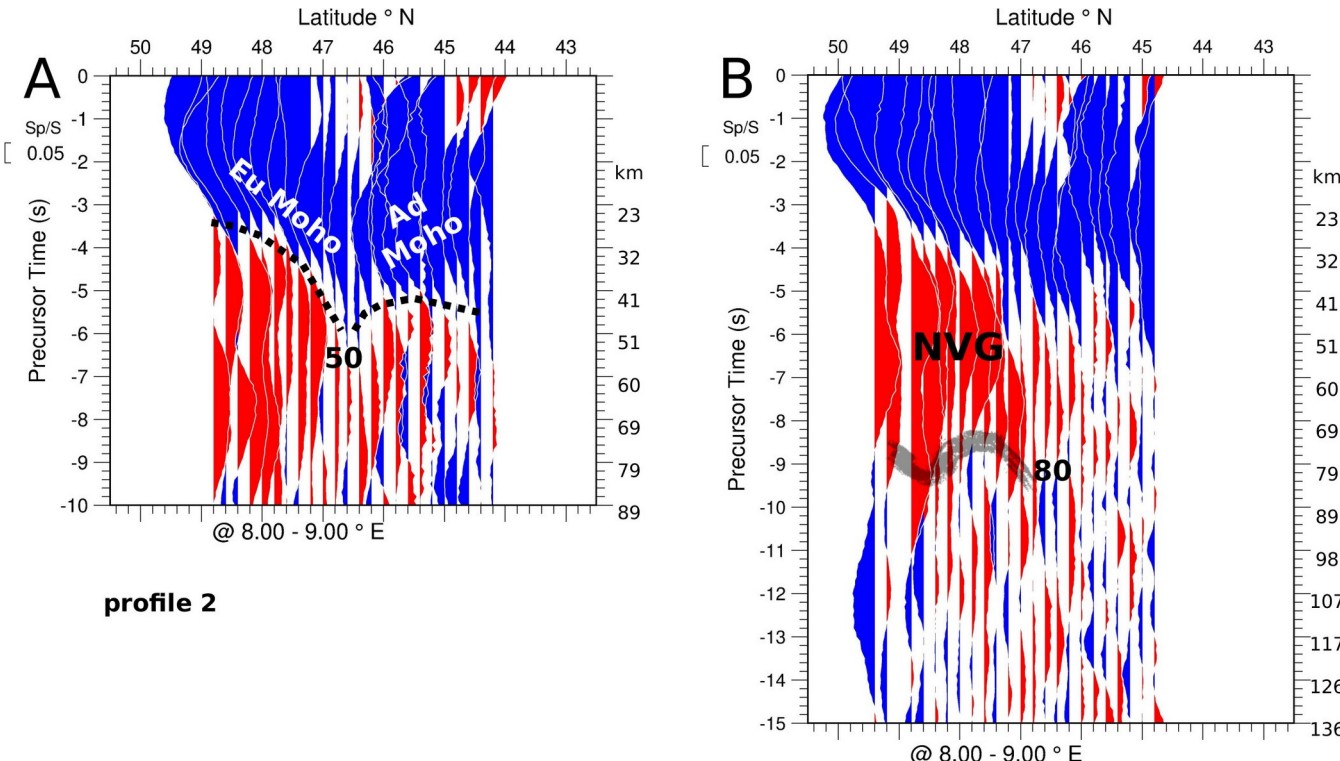

**Figure 4:** Profile 2: North-south profiles at 8-9°E. Note that the deepest point of the Moho at 46.5° latitude is slightly north the Europe-Adria boundary at around 46° according to Spada et al. (2013) (see Fig. 2).





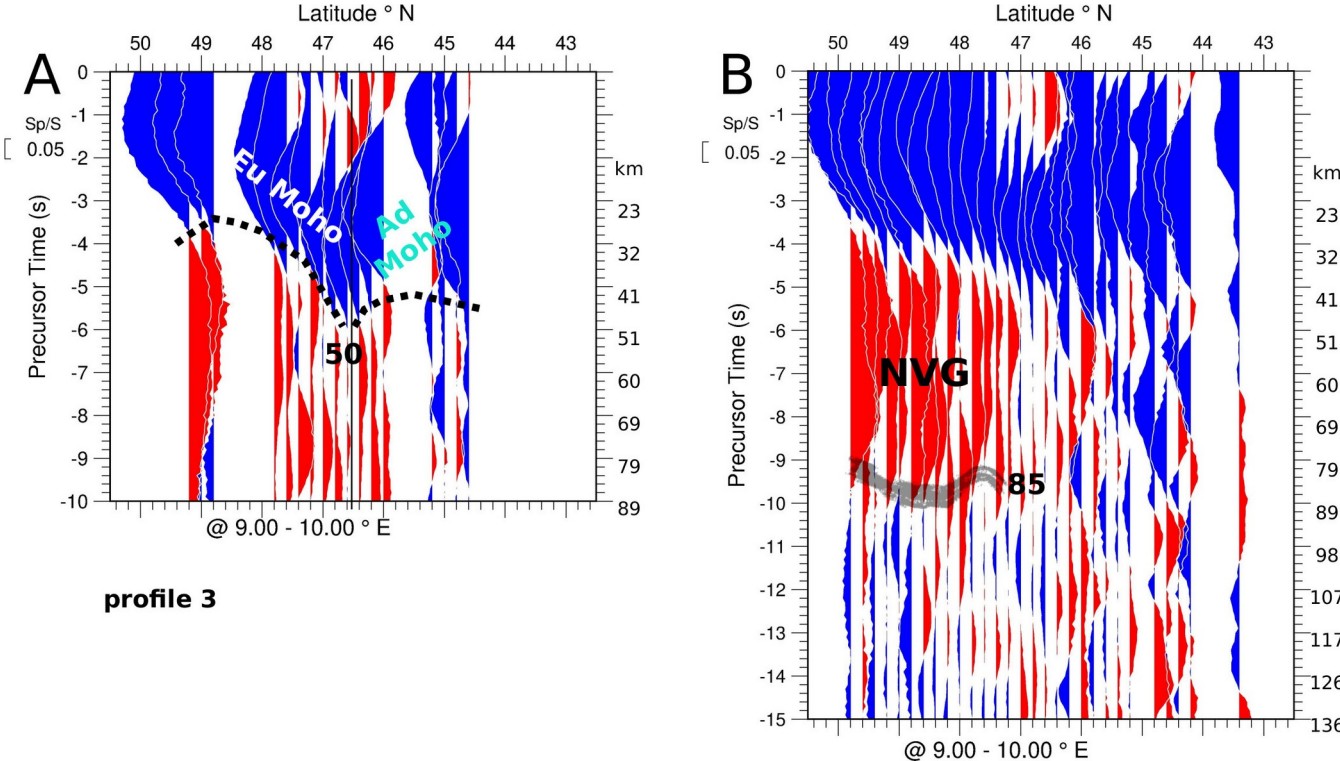

**Figure 5:** Profile 3: North-south profiles at 9-10°E. Note that the deepest point of the Moho at 46.5° latitude closely coincides with the Europe-Adria boundary according to Spada et al. (2013) (see Fig. 2).




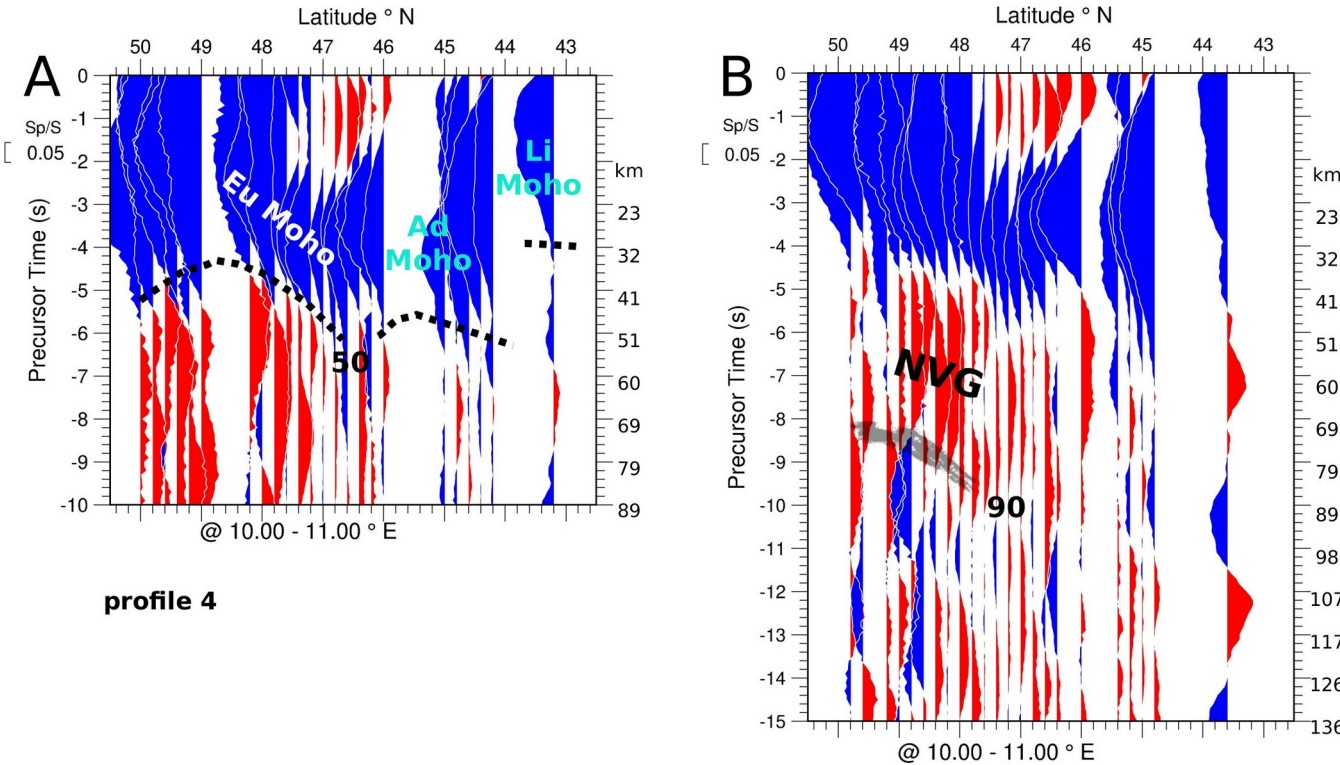

**Figure 6:** Profile 4: North-south profiles at 10-11°E. Note that the deepest point of the Moho at 46.5° latitude closely coincides with the Europe-Adria boundary according to Spada et al. (2013) (see Fig. 2). The base of the much shallower Ligurian Moho depicted in Fig. 2 marked "Li Moho" is visible at around 43.5° latitude.




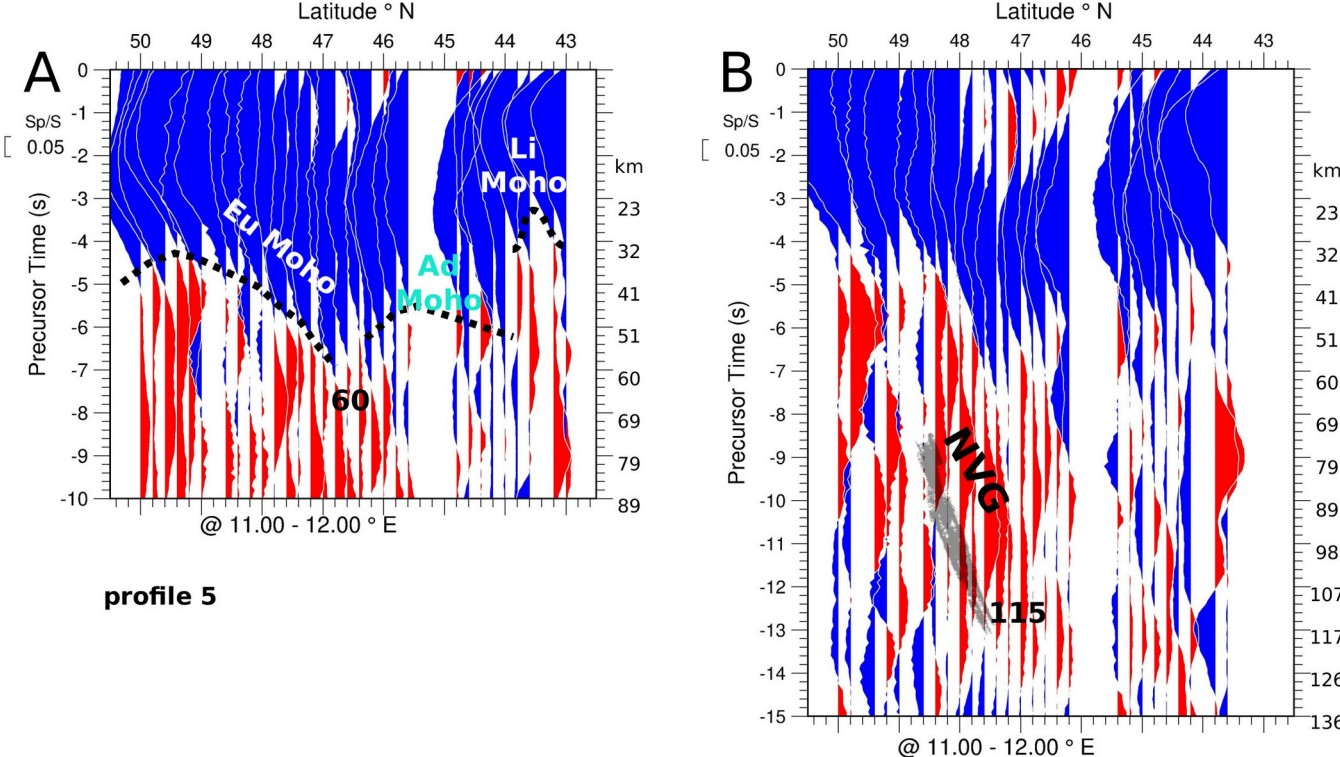

**Figure 7:** Profile 5: North-south profiles at 11-12°E, i.e. near the western margin of the Tauern Window. Note that the deepest point of the Moho at 46.5° latitude closely coincides with the Europe-Adria boundary according to Spada et al. (2013) (see Fig. 2). The base of the much shallower Ligurian Moho depicted in Fig. 2 marked "Li Moho" is visible at around 43.5° latitude.





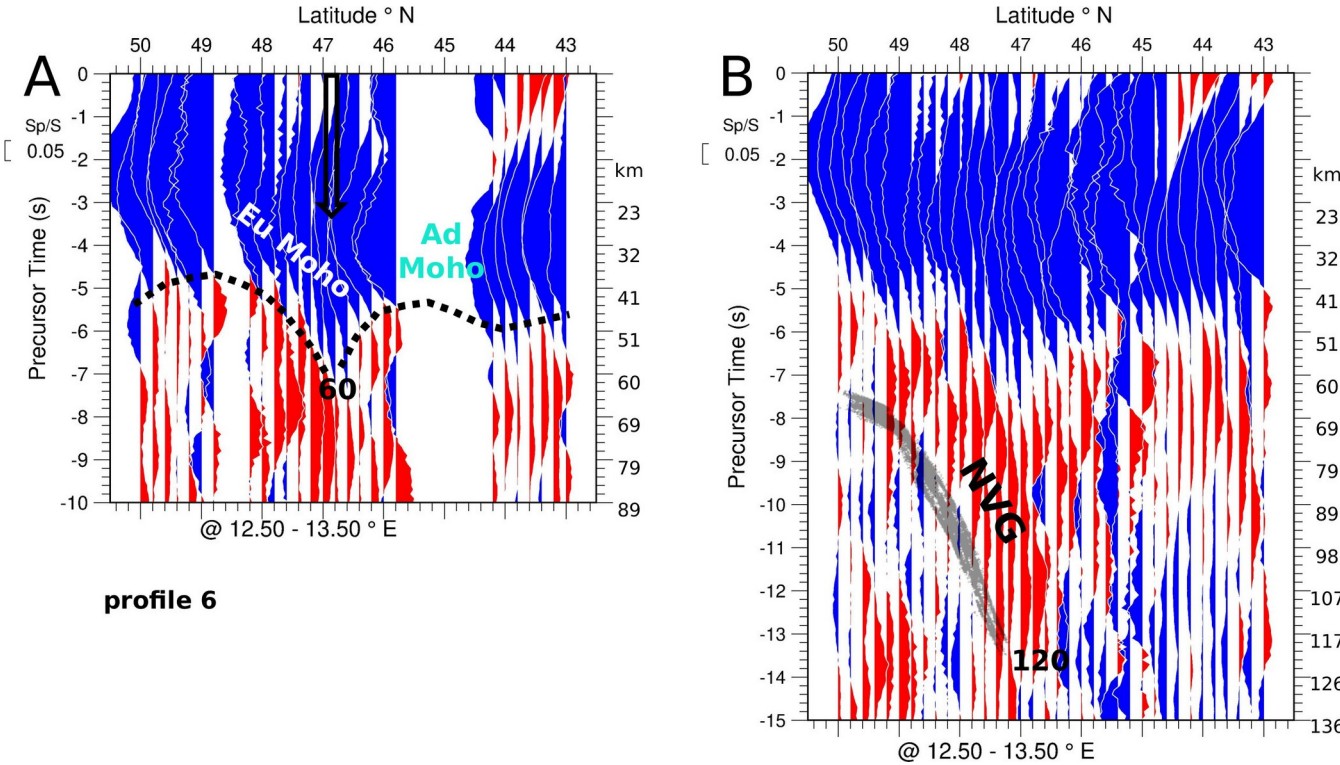

**Figure 8:** Profile 6: North-south profiles at 12.5-13.5°E, i.e. across the eastern part of the Tauern Window. Note that the deepest point of the Moho near 47° latitude, marked with a black arrow, closely coincides with the Europe-Adria boundary according to Spada et al. (2013) (see Fig. 2). This profile corridor does not reach the Ligurian Moho in the south.





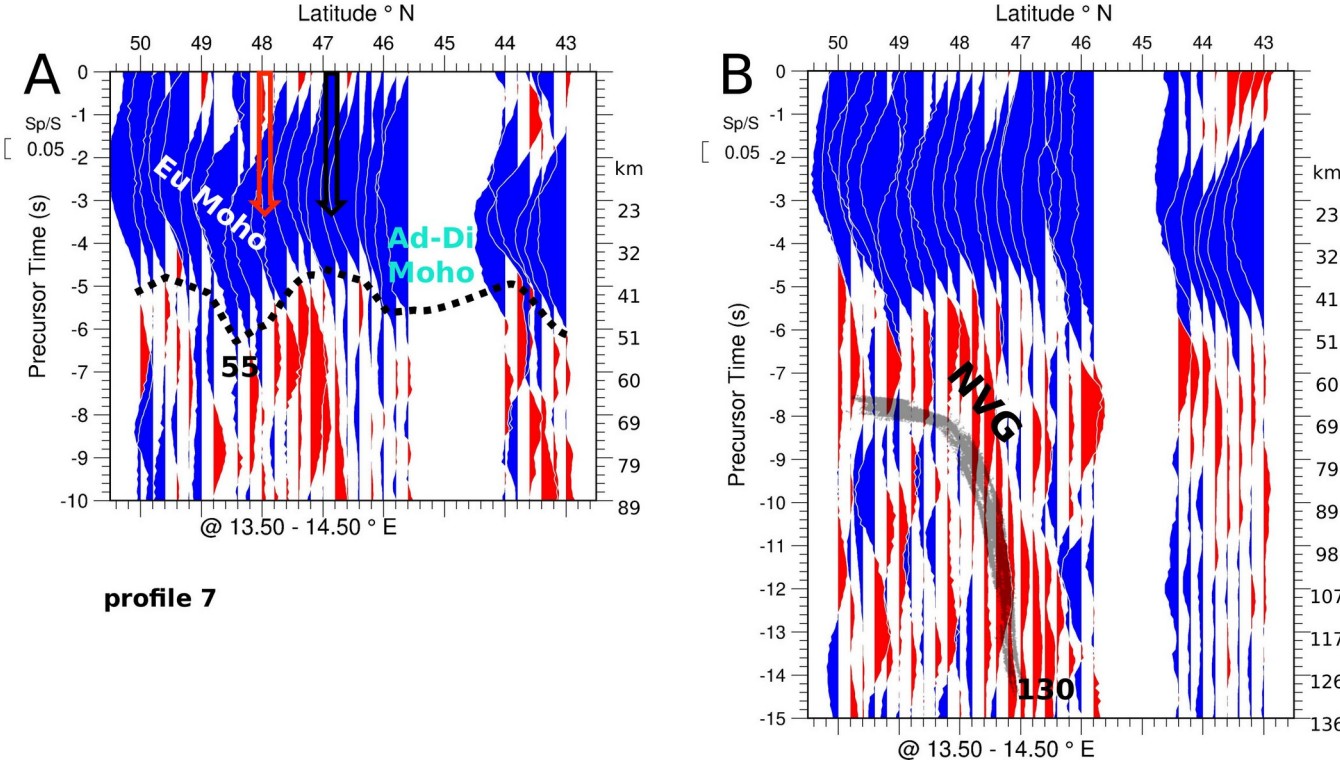

**Figure 9:** Profile 7: North-south profiles at 13.5-14.5°E, located easterly adjacent to the Tauern Window. "Ad-Di Moho" indicates Adriatic-Dinarides Moho. A) The black arrow locates latitude near 47° indicating where the Moho reached the deepest point in profile 6 of Fig. 8. Note the abrupt along strike change in terms of the Moho topography near 13° E longitude, where this section shows the exact opposite, namely a culmination rather than a through at this same latitude (compare with Fig. 8). This location coincides with the white area in the map of Spada et al. (2013) denoting an area where the Moho is ill constrained by their data ("Moho gap", Hetényi et al. 2018b). The red arrow indicates the location of the frontal thrust of the Alpine orogen.




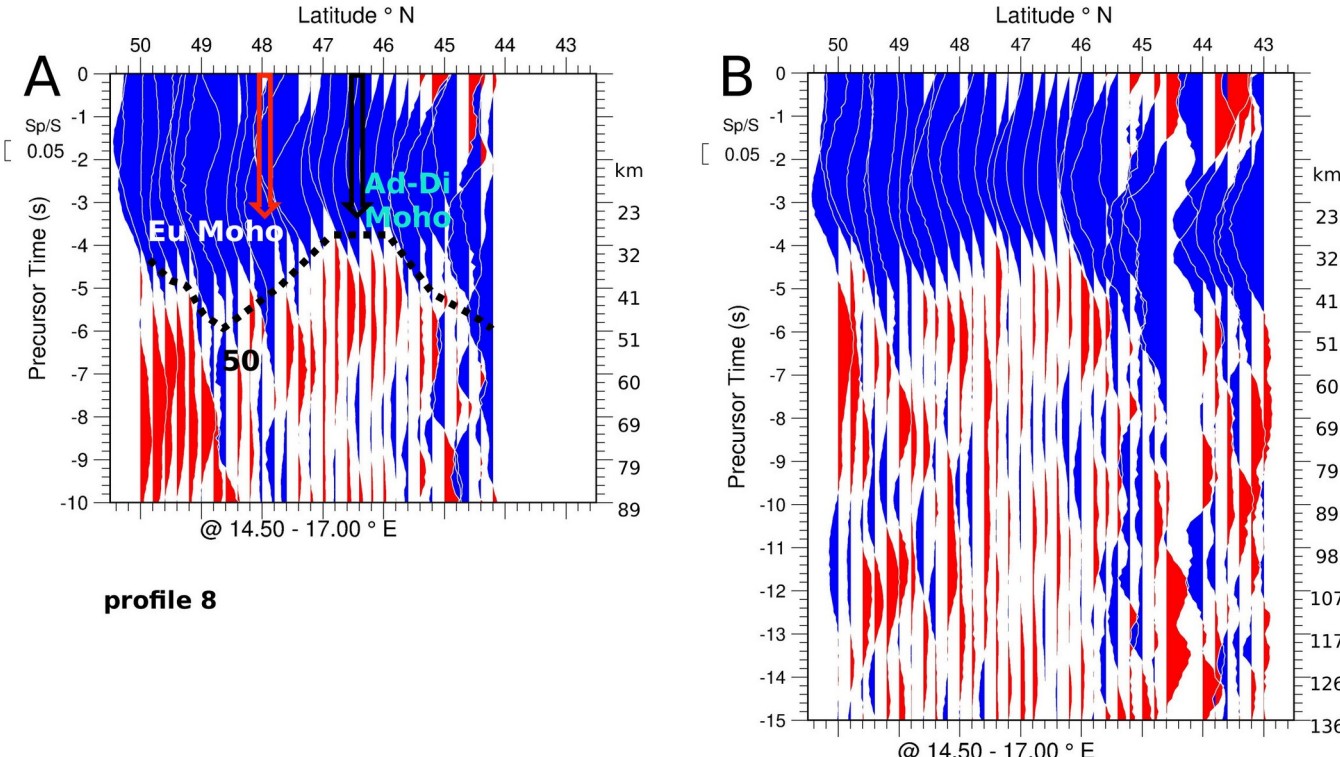

**Figure 10:** Profile 8: North-south profiles at 14.5-17°E. As in profile 7 of Fig. 9, the Moho culminates near the junction of European and Adriatic Moho at latitude 46.5° (i.e. at the Mid-Hungarian Fault Zone marked with a black arrow). Ad-Di denotes the Moho beneath the undeformed Adria plate dipping to the NE beneath the Dinarides. The red arrow indicates the location of the frontal thrust of the Alpine orogen.




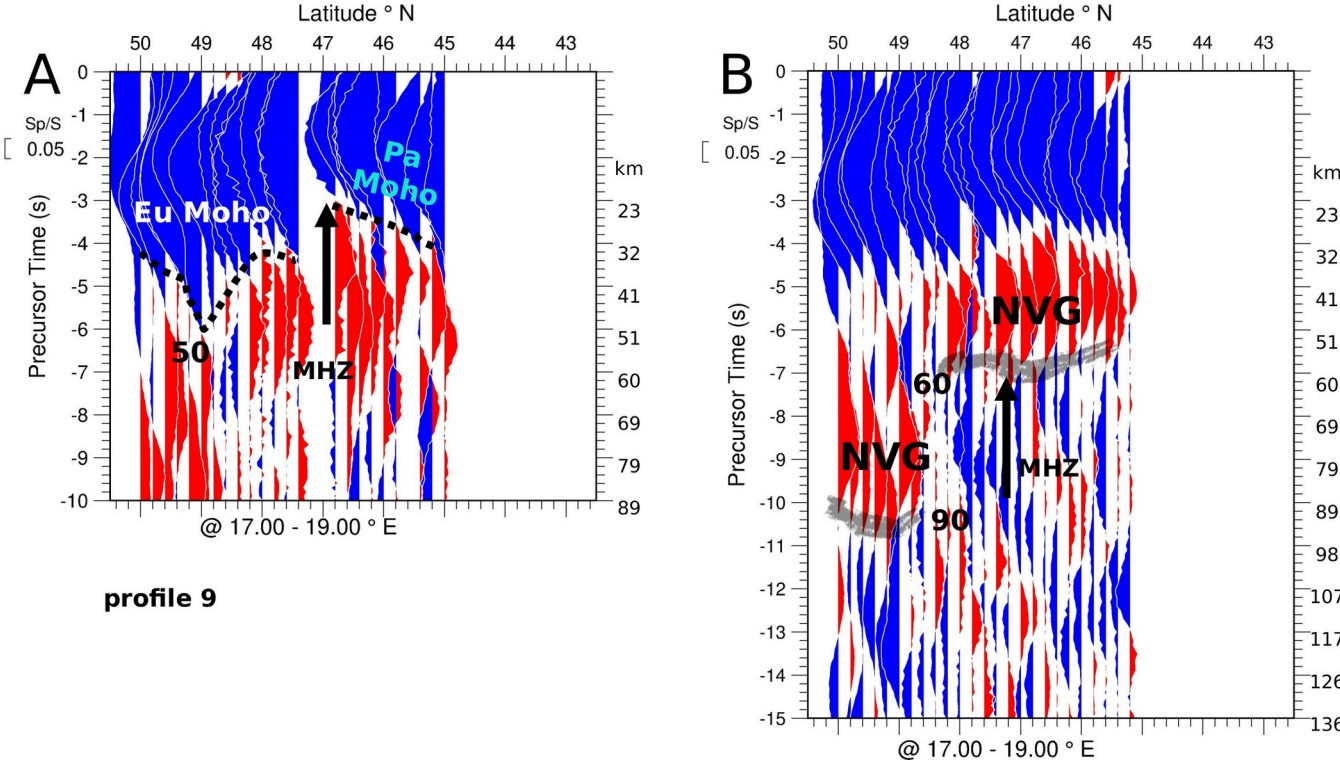

**Figure 11:** Profile 9: North-south profiles at 17-19°E. As in the profiles 8 of Fig. 10 the Moho culminates in the area of the Mid-Hungarian Fault Zone (MHZ) marked with a black arrow. In this profile the MHZ denotes the boundary of the substantially thinned European Moho to the shallow Moho of the Pannonian Basin south of Lake Balaton (marked Pa Moho).




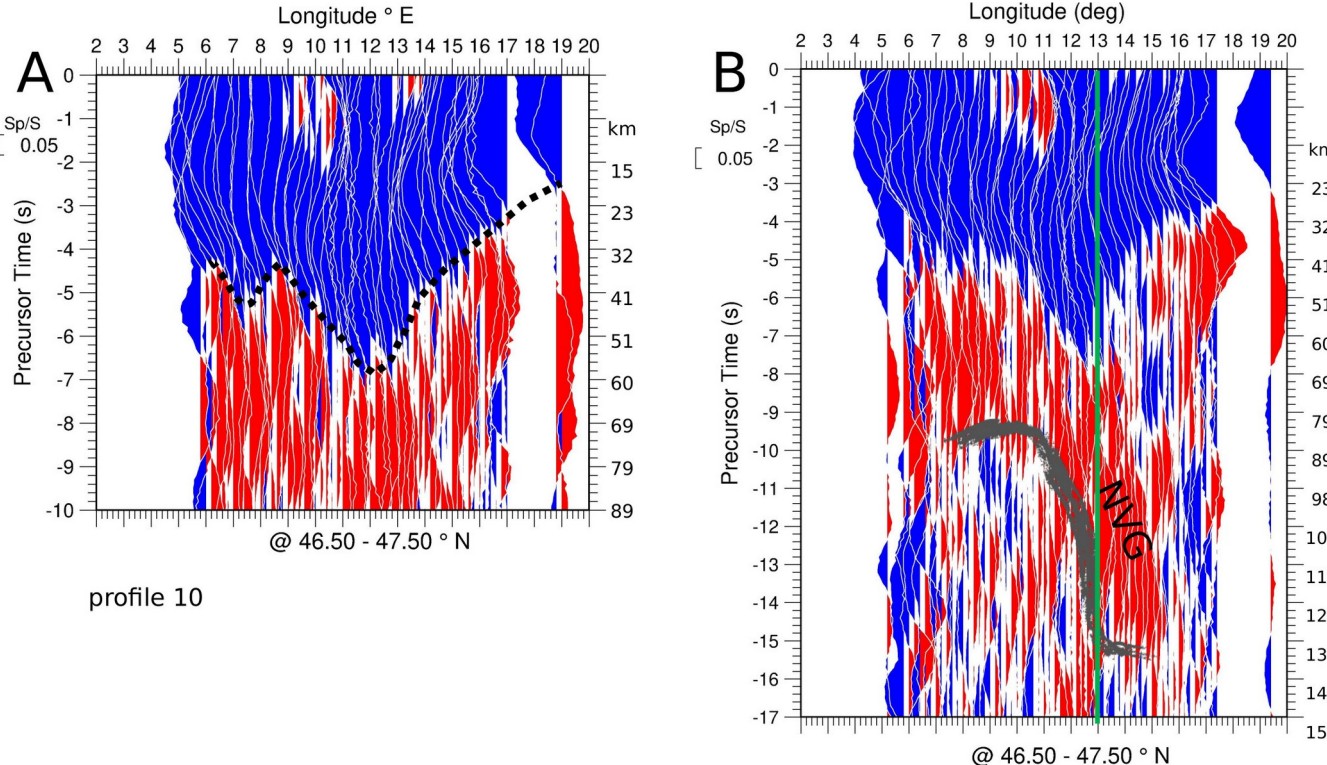

**Figure 12:** Profile 10: East-west profiles at 46.5-47.5°N, largely running within the European plate and straddling the plate boundary east of 11.5°E. Note vertical exaggeration of around 13 times in this E-W profile and those of Fig. 13 and 14. A) Approximate onsets of Moho signals along this profile are marked by a black dotted line. B) Grey shadow indicates the base of the NVG; green vertical line marks a very pronounced along-strike change at around 13°E discussed in the text.




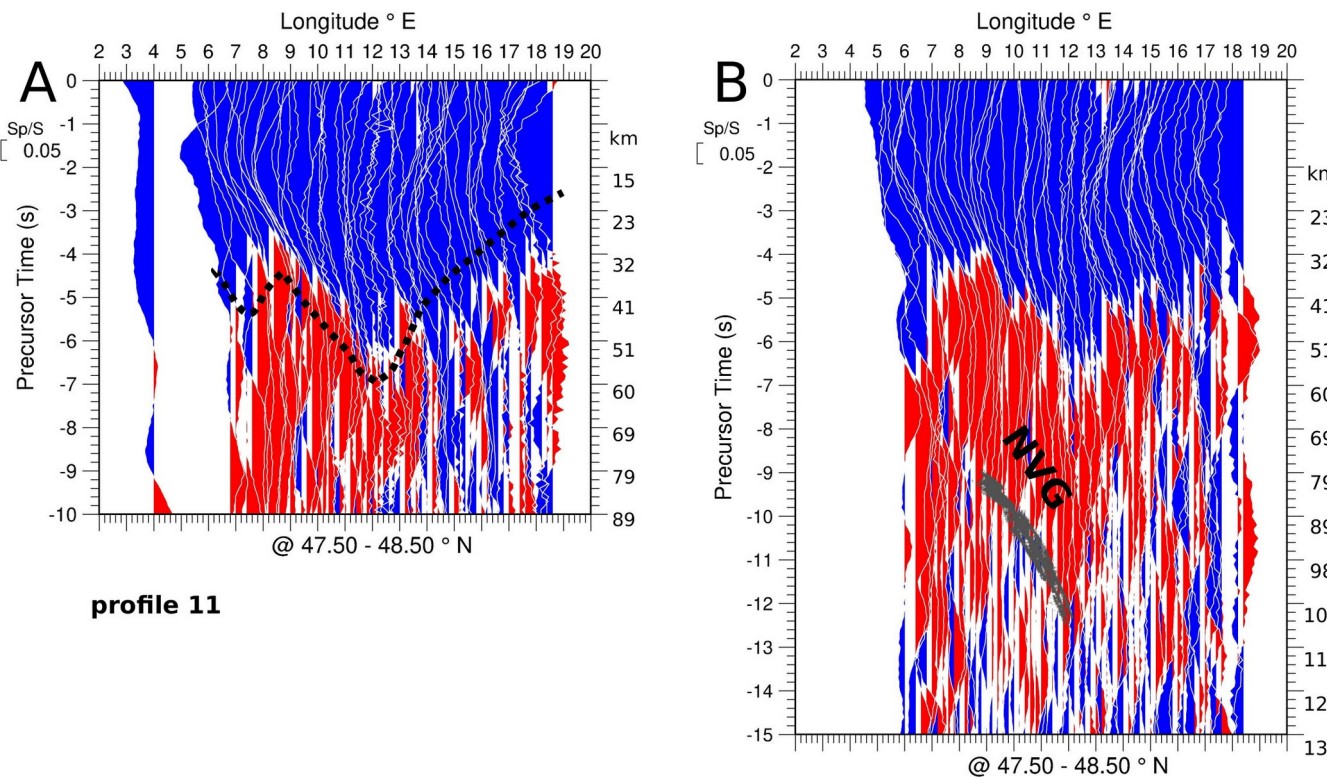

**Figure 13:** Profile 11: East-west profiles at 47.5-48.5°N running entirely within the European plate. A) The same black dotted line as in profile 10 (Fig. 12) is copied into this figure for comparison.






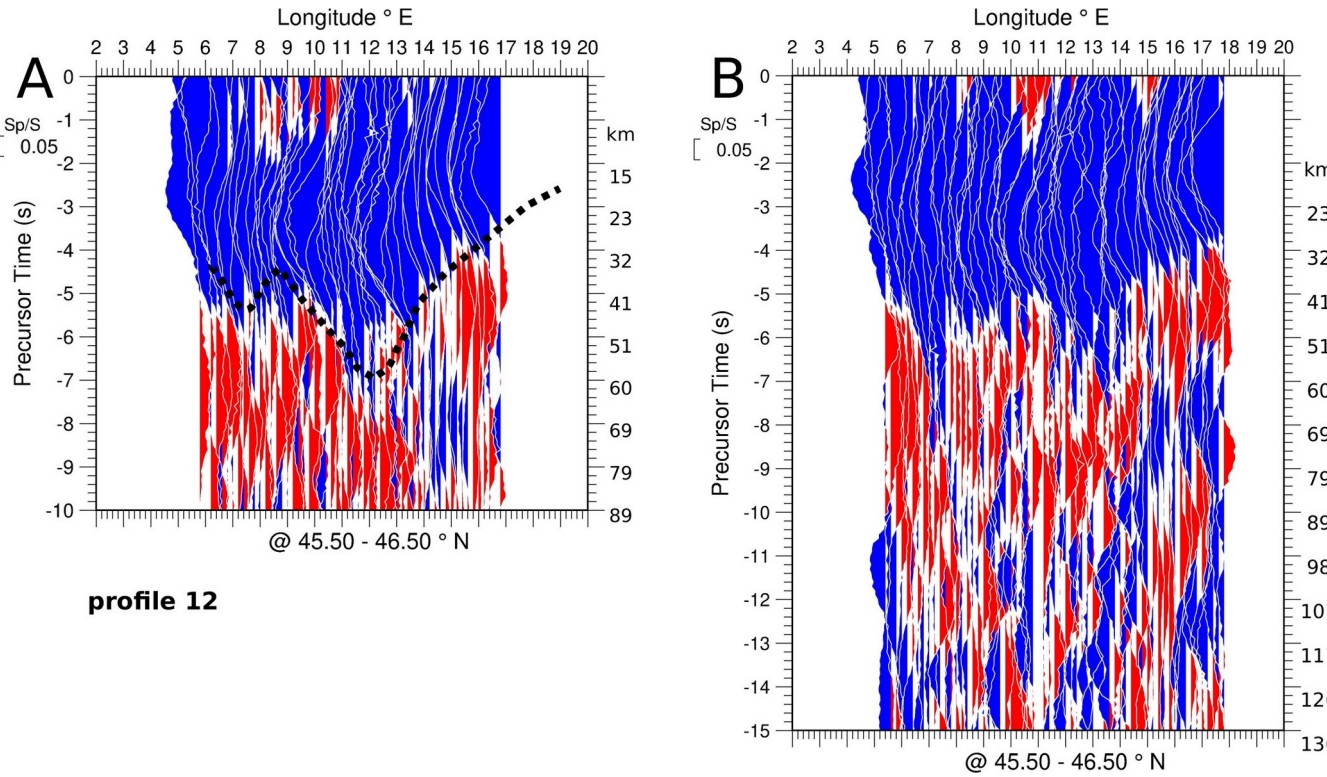

**Figure 14:** Profile 12: East-west profiles at 45.5-46.5°N, running within the Adriatic plate east of 8°E. A) The same black dotted line as in profile 10 (Fig. 12) is copied into this figure for comparison.






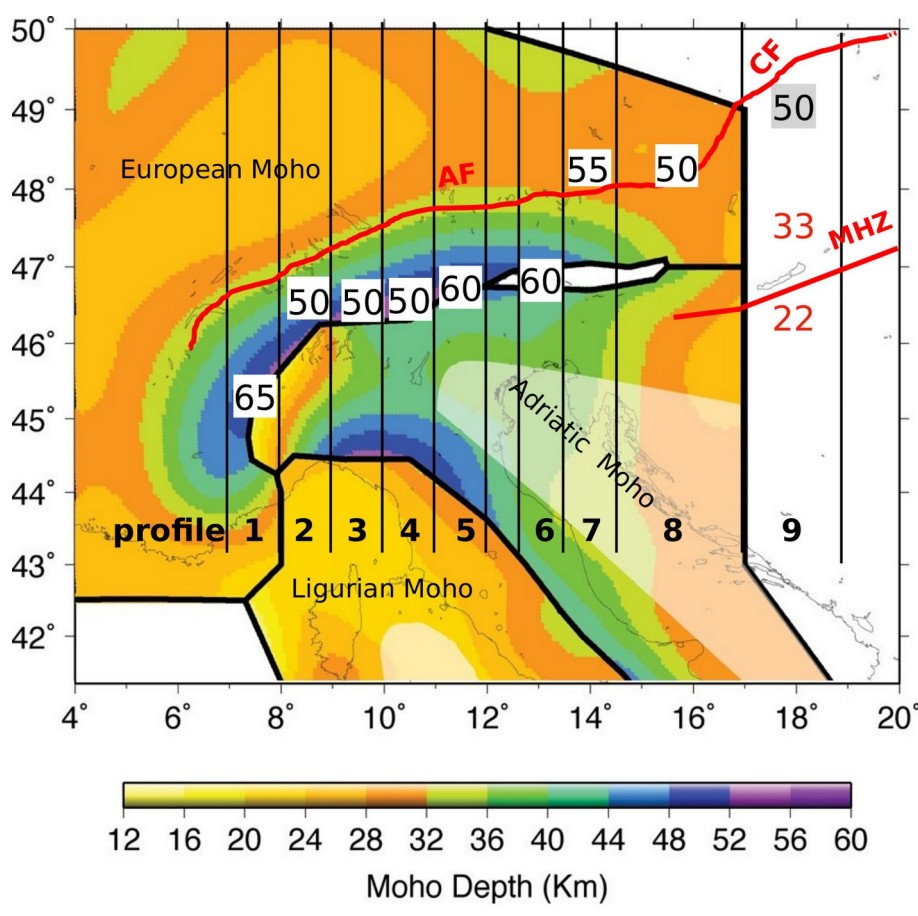

**Figure 15:** Moho map of Spada et al. (2013) indicating the maximum recorded depths of the Moho along the Alpine chain west of 13.5°E and below the southern part of the Bohemian Massif and below the northernmost West Carpathians, marked in kilometers. Numbers are taken from profiles 1-12 A. In addition the Moho depths north and south of the Mid-Hungarian Fault Zone (MHZ) are marked in red characters. AF, CF and MHZ mark Alpine Front, Carpathian Front and Mid-Hungarian Fault Zone, respectively.






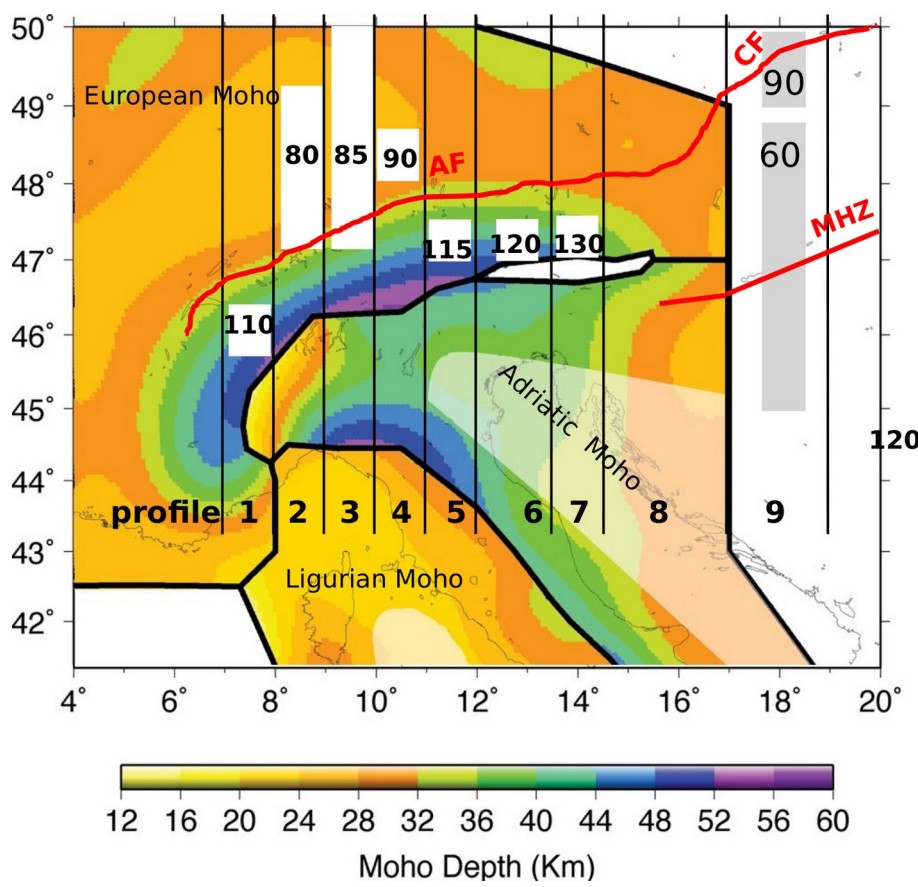

**Figure 16:** Moho map of Spada et al. (2013) indicating the maximum recorded depth of the signals at the base of the NVG as marked in profiles 1-12 B. Small white bins in the background of the numbers indicate steeply dipping NVG. Larger backgrounds indicate nearly horizontal NVG. Cyan quadrant marks the region with relatively good NVG observations, coinciding with the area of the "Moho gap" (white).



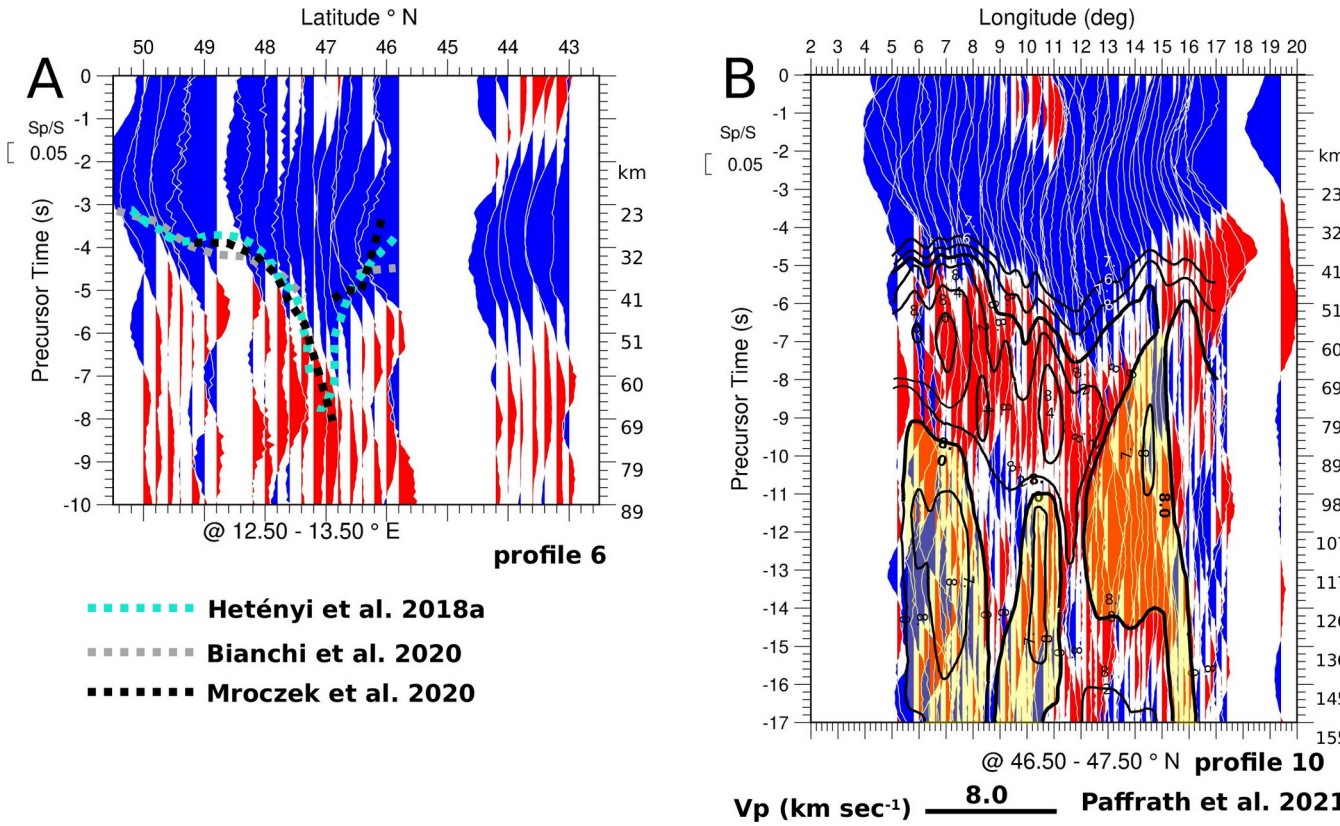

**Figure 17:** Copies of profiles 6 and 10, superposed with results of earlier studies for comparison. A) Comparison of profile 6, focusing on Moho depth, with profiles of Hetényi et al. (2018b) based on P receiver functions, Bianchi et al. (2020) based on PKP crustal multiples and Mroczek et al. (2020) also based on P receiver functions. We did not mark Moho onsets in this figure as in Fig. 8A to allow comparison with the original data without our interpretation. B) Comparison of profile 10, focusing on the depth of the base of the NVG indicated by a grey shadow in Fig. 12B, with absolute P-wave velocities obtained from teleseismic P-wave travel time tomography by Paffrath et al. (2021).





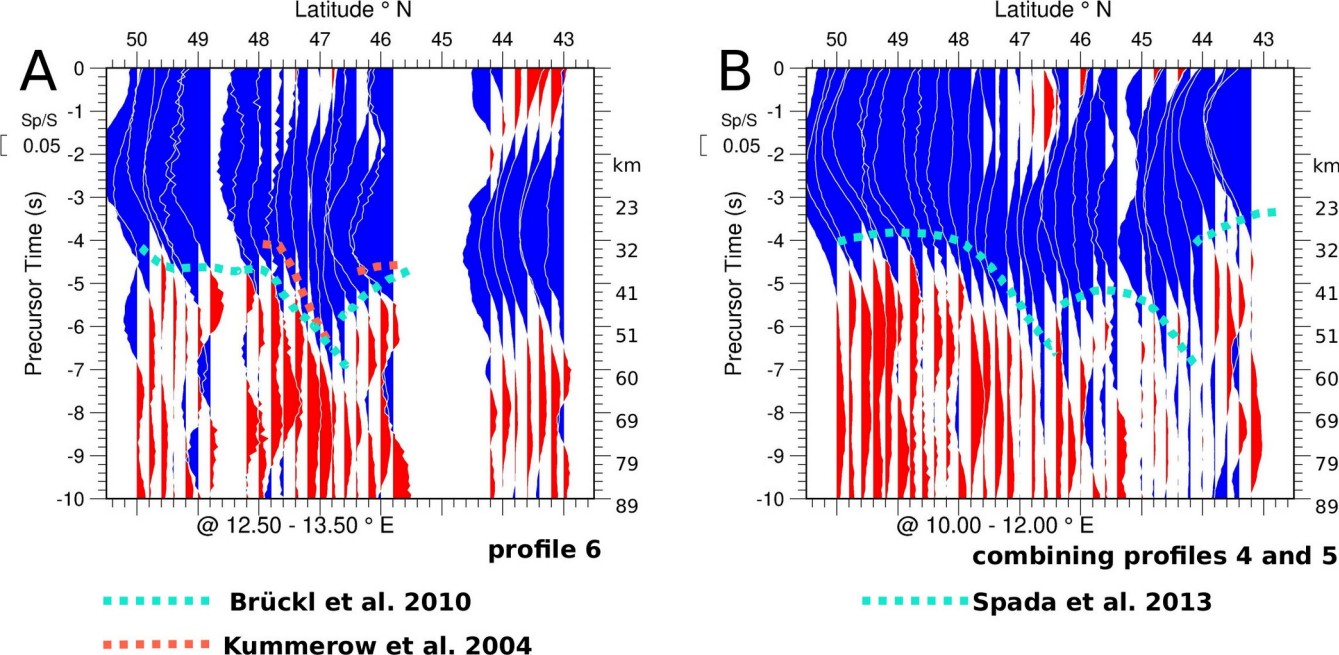

**Figure 18:** Copies of north-south profiles with results of earlier studies for comparison. A) Comparison of profile 6 (Fig. 8A) with Brückl et al. (2010; their Fig. 3a), who used Moho data of Behm et al. (2007) based on wide-angle reflection and refraction data, and, with Kummerow et al. (2004) using the P receiver function method. B) Comparison of a combination of the corridors of profiles 4 and 5 (Figs.6A and 7A) with the same profile taken from map of Spada et al. (2013).




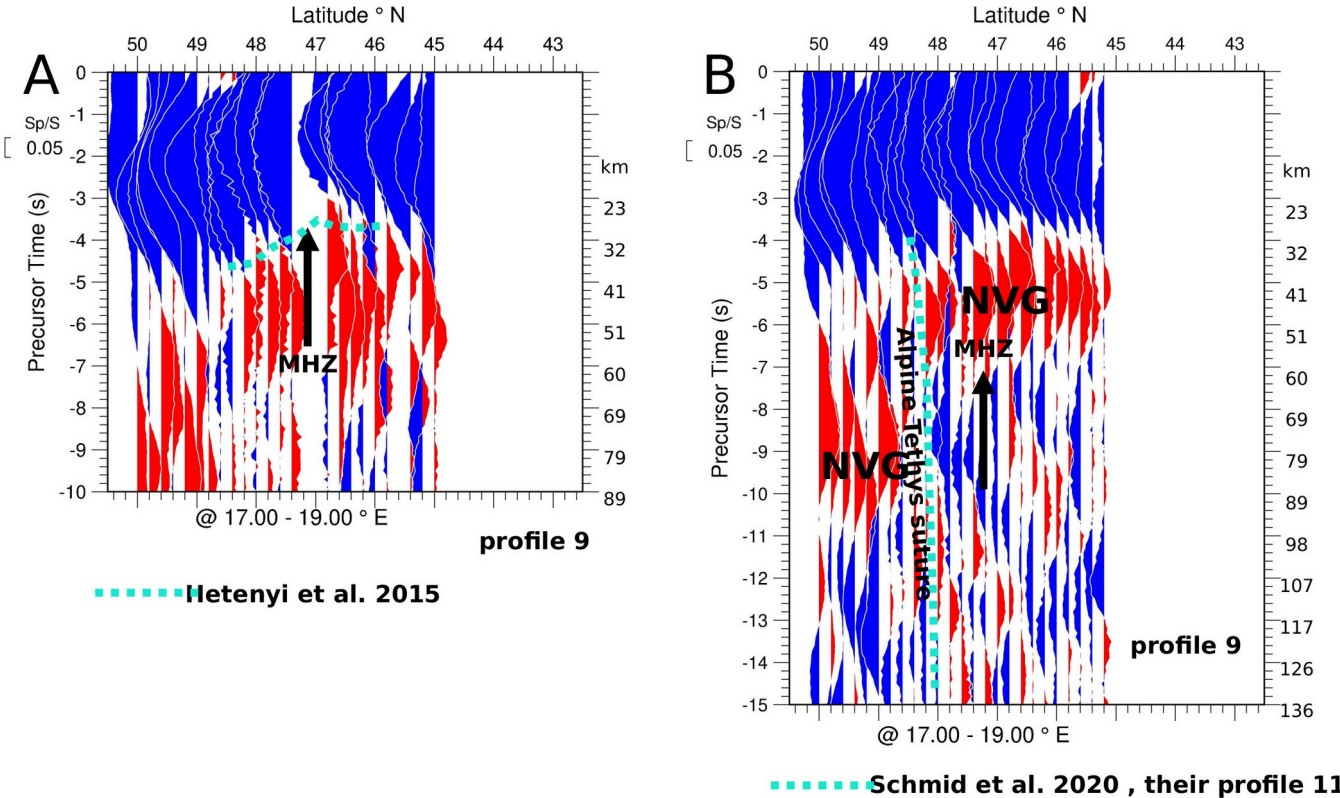

**Figure 19:** A) Copy of profile 9 (Fig. 11A) across the MHZ superimposed with a study by Hetényi et al. (2015) who used P receiver functions. B) Copy of profile 9 (Fig. 11B) superimposed with the location of the Alpine Tethys suture as interpreted by Handy et al. (2021) based on mantle tomography by Paffrath et al. (2021).






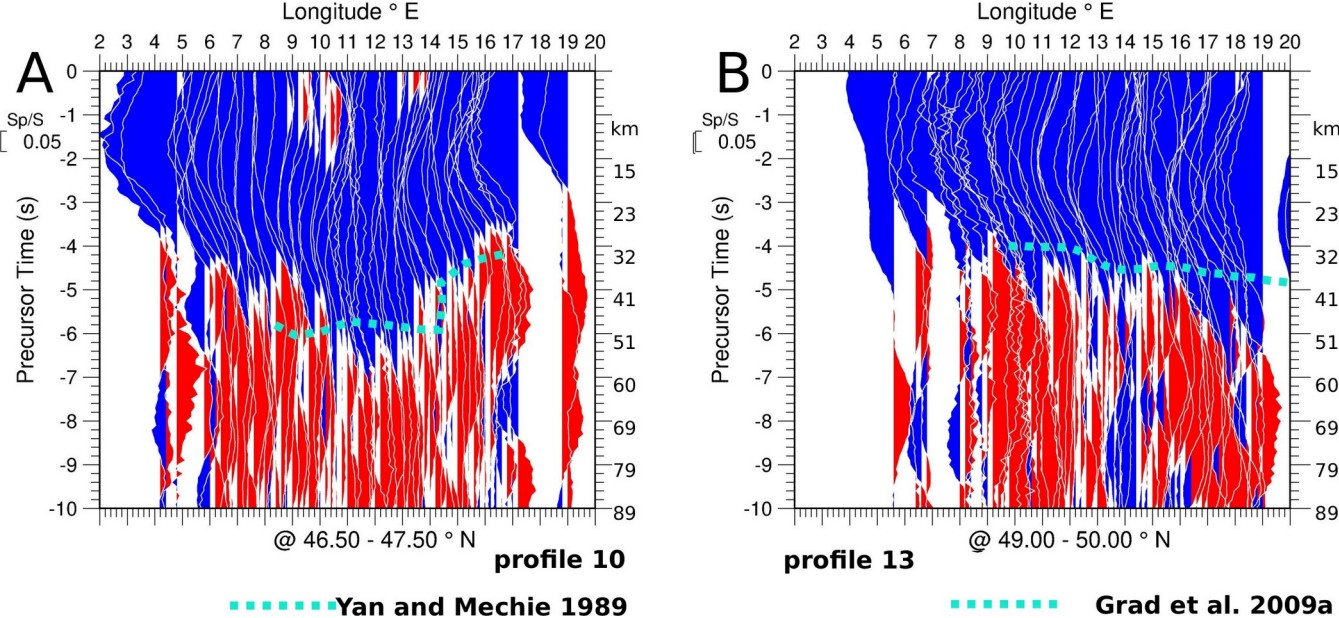

**Figure 20:** A) Copy of profile 10 (Fig. 12A) with results of Yan and Mechie (1989) who used data from a refraction profile along the Alpine chain. B) Additional east-west profile well north of the earlier profiles shown, constructed for comparison with the Moho map of Grad et al. (2009a) and comparison with profile 10 of Fig. 20A. Note the substantial difference in Moho depth between our data east of 14°E and those of Grad et al. (2009a). Also note that we see pronounced Moho deepening towards the east in this northerly located profile located in the European lithosphere, which contrasts with Moho shallowing towards the east in the southern profile of Fig. 20A located in the Pannonian region (see Fig. 2 for location of the profiles).