# Peer review of "Moho and uppermost mantle structure in the Alpine area from"

_Solid Earth, 2021_

## Author Response (AR1)

**Response to the Reviewers**

Author's response
Author's changes in the manuscript
Reviewers comments

- RC1: 'Comment on se-2021-33', Anonymous Referee #1, 07 May 2021  reply This work is based on a large amount of data coming from experiments within the framework of the AlpArray experiment (AlpArray, SWATH-D) and permanent networks, and the application of a novel method using S-to-P conversions ('causal' SRF or C-RF). The objective was to investigate the Moho and lithospheric structure of the greater Alpine region. The size of the analized area allows for an overall view of this region and the general trends along-strike of the Alps, although the paper focuses particularly on the Tauern window, a key area, and the change that occurs across 13°E all the way to the Pannonian basin. The paper is well written, the comparison with previous studies interesting and the results clearly stated.

  In regard to the observations, it is nice to see the stacked wave-forms in the profiles but it is hard to understand how the discontinuities are plotted from the first arrivals. In fact, in general the choice of the line representing the discontinuity often seems arbitrary. Even when the choice of the converted phase "bump " is clear it might be difficult to ascertain where the line representing the discontinuity should be placed, particularly when there is an emergent "bump". This is especially true for the small signals converted below the lithosphere, that are identified as NVG (negative velocity gradient). It seems that it is possible to identify some clear patterns in the profiles but in many cases it is not so. For example, below-Moho Profile 9B (Figure 11) is convincing, it really shows a jump in the NVG across lat 48.5°, but other profiles are very noisy, for example the discontinuity traced for profile 1B (Figure 3) seems arbitrary. Also, in profile 4B how can you trace such smooth discontinuity when the arrivals are jumping up and down? In profile 5B one cannot really see the arrivals, amplitude is too small, they are flat.....For this reason the suggestion is to have a more cautious approach in identifying the interfaces where S-to-P takes place, and state clearly which are the reliable observation. Some question marks would be appropriate on the figures in dubious cases or when there is more than one choice. This is particularly the case for the B part of the profiles. It seems that the authors are aware of these uncertainty since in the conclusions they only summarize the observations that are more reliable.

- These comments are fair and understandable. We have tried to make clear that the NVGs are areas of scattered small-scale converters which do not form a single clear "laterally homogeneous discontinuity", which could perhaps be interpreted as LAB. Comparable observations in the mantle lithosphere are rare and therefore any new information is important. The special features of the NVG observations confirm doubts about the existence of a clear seismic LAB, at least in the Alpine area. We will express this more clearly and mark it also more clear in the figures. These signals are also very weak with amplitudes only about 2-5% of the incident signals. However we think they are still important indicators of regions where the velocity is decreasing downward. The fact that the NVGs are concentrated in some regions permits their interpretation, together with other data, as possible indicators of significant geodynamic processes We did not yet pick Moho onsets at the stacked traces because of apparent precursors of the Moho onsets. We are working on that problem by comparing waveforms of SV signals with Moho waveforms. This should help to determine Moho onsets more reliably and will allow to produce a map of the Moho depths. The same applies for the NVG signals. The dashed lines near the beginning of the Moho onsets do not mark Moho onset times. They just mark relative changes of the Moho depths along the profiles.

- We have explained in more detail that the lines before the Moho and NVG signals are not marking accurate arrival times. They are intended to mark mainly lateral changes of the discontinuities. Determining accurate arrival times requires wave form comparisons, which will be done in a future study. In several figures we made adjustments of the lines to better fit the signals better.

- Finally, filtering and any additional processing of the waveform can cause unwanted effects that might lead to misintepretation. On the other hand, not filtering can also lead to problems due to unsupressed noise which might also lead to misintepretation. The strenghts and weaknesses of both approaches should be pointed out.

- We used only broadband signals with high signal-to-noise ratio of SV. Frequency filtering can therefore not improve the signal-to-noise ratio in the range of the main periods. The advantage of filtering is the improvement of the signal-to noise ratio for weak signals. One of our goals in an upcoming manuscript is the interpretation of the waveforms of the Moho and possibly the NVG signals, which would be distorted by filtering.

- We added a statement that we did not use filtering (including deconvolution) because we intend to interpret the original waveforms of the converted waves in a later study.

Specific *comments* (line numbers on the left)

59-60 "corrected for the sign of the onset", does it mean that negative phases are multiplied by -1 so all SV bumps are positive? YES

-

- 67-68 data selection 50% noise, it is ok for the Moho signal (~10% amplitude) but perhaps not for deeper conversions (~few%). It seems that this is a key aspect determining if sub-Moho signals can be detected. Perhaps in such a heterogeneous area (high signal generated noise) the threshold on the noise on the P component should be lower. It is explained in the text that to have enough waveforms this threshold cannot be too small, but another way to increase the number of wavefomr in each cell is to increase the cell size.To increase the S/N, especially for the sub-Moho part, it might be necessary to loose some (hypothetical) resolution.

- *We agree that permitting a 50% noise limit seems high for the very small NVG signals. This number results from experiments, which we have done. Increasing the cell size would, of course, improve the situation, but the price would be the lateral resolution. We think we found a good compromise.*

- This is discussed in the text, there is practically no signal generated noise before the SV signal.

- 94 "The signal forms of the Moho (and other) conversions are determined mainly by the signal forms of the incident SV signals." Is this the way the curves are identified in the profiles? If yes, this is also a key aspect, stated this way it is vague. Please explain more clearly, possibly with an example (maybe a figure as Supplementary material).

- *We need to make this clearer. In case of a single discontinuity, the signal form of the converted wave is determined by the signal form of the incident wave. However, if there are several discontinuities close to each other, the resulting signal form will be more complicated. In case of the Moho signal, a comparison of both signals (SV and Moho signals) should be very meaningful. We are now working on that question in a subsequent manuscript also presenting a Moho map. We hope to determine the Moho arrival time more accurately this way. In the present manuscript we did not determine Moho times (with the exception of the depth estimates of the largest Moho depths shown in Fig. 15). In many cases it seems difficult to determine the Moho arrival time because of possible precursors, which could indicate sub-Moho conversions with the same sign. By the way, this is one of the reasons why we avoided deconvolution. Deconvolution changes the waveforms and makes it very likely difficult to identify small precursors. The dashed lines in the Figures A mark only roughly the general trend of the Moho signals along the profile.*

- *This is discussed in more detail in the revised manuscript.*

- *97-99 How can we be sure that the negative phases below the Moho that make up the NVG are real features and not part of the intereference pattern between the "real" phases or an effect of the interference of signal generated noise ? Is there a way to determine the signficance of these scattered negative bumps ?*

- *The signals we called NVG are the first arriving signals and we did not see nor do we expect larger signals before the NVG signals which could generate noise. However, theoretically the NVG signals could generate noise, which could disturb the Moho signals. But comparing the amplitudes of both signals, this possibility seems insignificant.*

- *We added a statement claiming that no signal generated noise is expected. Such noise could practically only come from P multiples, which are weak and reduced by the stacking method.*

- 113-116 I agree for profiles 2-4, but for profiles 4-6 the Adriatic Moho between 45 and 46 is completely inferred since there is no data. For profiles 4-6 I think it is not possible (at least with these display) to see the culmination of the Adriatic Moho coming from the observations.

- *You are certainly right. We will change this, also in the figures.*

- The figures have been changed.

- 117-119 perhaps from the profiles since the culmination of the Adriatic Moho is seen in profiles 2-4 it would be safer to say change "at least west of 11°" --> "west of 11°"  *OK*  *done*

- Figure 3a. In Profile 1A (figure 3) it is not clear how is the Moho signal onset is identified on the waveforms. The dotted curve does not seem to fall on the first arrival of several traces, it seems it should be more wavy. In particular, the dotted curve does not seem to follow the data (origin time) between 46.5 and 45, in fact it is difficult to identify the arrival time. Similar comments can be applied to other Moho profiles. In general, a clear example on how this interface is identified should be shown.

- *The dotted line does not mark the Moho arrival times. As mentioned above, it marks only the trend of the Moho along the profile. Moho arrival times will be determined in an upcoming manuscript.*

- *Is more explained now.*

- 120-133 In Figure 9A it appears that the onset of the Moho signal under the black arrow corresponds to about 45-47km depth. Also, the waveforms of the dipping (European) Moho within about 48°-49° seem to interfere and cause the Moho signal to be very broad and extended at depth. It is very difficult to identify the onset on these very emergent arrivals. This is true also for other profiles, for example profile 8 in Figure 10A. Since this are key observations for the interpretation of a change of the subduction style across 13° the authors need to show how this interfaces are constructed from the data in a clearer and more convincing way.

- *This is a very significant point. It is a typical example for positive precursors of the Moho. The determination of the arrival time is therefore difficult. The comparison of the SV and Moho waveforms could help. We are working on this issue now and give therefore no Moho arrival times in the present paper (except some approximate values in Fig.17). We will improve the explanation of these circumstances in the text.*

- The discussion of this problem is extended and as a consequence the interpretation of the Moho across the MHZ is changed.

- 234-236 Figure 17B is very difficult to read. In particular, one cannot see the values on the lines with constant velocity of Paffrath et al. so the comparison with the present work is also difficult.

- *Thank you for pointing out that Fig. 17B is indeed very difficult to read. We improved this figure for clarity by increasing the size of the numbers denoting p-wave velocities*

- the figure was changed

- 250-254 From Figure 17B seems that the positive velocity anomaly gradient of Paffrath et al starts at 4° and continues up until about 11° and that the agreement with the present work is between 11° and 14°.

- *It appears that our message presented in the entire paragraph between lines 234-256 was misunderstood. We believe that this is probably due to the poor style of writing on our side. Hence we rewrote the same paragraph to enhance clarity and structure of the arguments. We hope that the point we make now find the acceptance of the reviewers.*

- *we wrote a new text*

- 259-260 It seems that, given the seemingly arbitrary choice of the Moho interface it is difficult to compare the two results. Perhaps, there seems to be an asymmetry but it should be more precisely shown.

  286-287 Can't see a dotted black line in Figure 20B

- *In Fig.18 (as in most other figures) we just displayed our waveforms without picking own Moho onsets. The depth determinations (dotted lines) from the other authors are relatively close to the onsets of our Moho waveforms. The in Fig.20B is cyan, not black.*

- *We* explained this better in the modified text.

- 317-321 It is true that filtering and any additional processing of the waveform can cause unwanted effects that might lead to misintepretation. On the other hand not filtering can also have problems due to unsupressed noise which might also lead to misintepretation. The strenghts and weaknesses of both approaches should be pointed out.

- *this question is repeated from above*

-

-

-  **RC2**: 'Peer-review of ms. SE-2021-33 by Kind et al.', Anonymous Referee #2, 01 Jun 2021  reply Peer-review of ms. SE-2021-33 "Moho and uppermost mantle structure in the greater Alpine area from S-to-P converted waves" by Kind et al.

PAPER AND REVIEW SUMMARY

The manuscript presents a large amount of teleseismic data of P phases preceding direct S arrivals, to deduce the crustal thickness and sub-crustal velocity structure. While this method is typically used with deconvolution and for targeting the lithosphere-asthenosphere boundary, the Authors argue for a direct use of raw waveforms with almost no additional processing. The methodology is of interest but further information on the details are needed, as currently the method is not reproducible. Moreover, due to some simplifications, uncertainties weigh on the results, therefore these need to be quantified. The results bear on the Moho depth, with comparison to earlier studies, as well as negative velocity gradients (NVG) below the crust. Some of the interpretation is sound but some others seem to be subjective, hence more caution would be wise. For example, among the three points presented as main conclusions: (1) the offset across the Mid-Hungarian Zone is not supported by the data and contradicts all previous results using more appropriate methodology, therefore it does not appear well-resolved and credible; (2) the Moho trough in the Bohemian Massif connecting with the Western Carpathians is a nice and somewhat new result; (3) the interpretation of NVGs is very subjective and other Readers may find more or less of the interpreted results, I would use more caution, and possibly focus on amplitude variations.

The text itself is well written for the language, and figures are technically clearly presented.

Overall, the study is worth being published in Solid Earth, however moderate additions to methodology and major revision of the interpretation is appropriate. Further details are provided below with line numbers referring to manuscript lines and figures as in the preprint.

- *This paper is not a paper on a new method. It uses a modification of the classical S receiver function method described earlier (Kind et al. 2020). However, we will expand comments on the method. We agree with the reviewer that the question of the Moho across the Mid-Hungarian Zone is not very well presented in our manuscript and we will renew that. We will also improve the discussion on the Moho trough below the southern edge of the Bohemian Massif, which requires careful waveform analysis. We will also better explain the differences of our interpretation of the Moho and the NVG signals.*

- We added some additional explanations regarding the method.

- We thank the reviewer for his critical comments on the Moho at the MHZ. We have completely rewritten this part. We agree now with earlier observation on the Moho structure at the MHZ. The apparent differences in Moho depth may also be explained by a second weaker discontinuity below the actual Moho north of the MHZ.

POINTS OF MORE IMPORTANT CONCERN PRIOR TO INTER-PRETATION

1. The method presented by the authors is suitable for imaging smooth, broad spatial-scale structure of the lithosphere, and not really for particular local details or sharp variations. Both inherently due to S waves, and because of the selected profile projections and widths. This should be declared clearly in the ms. This is especially important as the comparison with earlier results using inherently higher-resolution data or/and methods needs to reflect the ability and limitations of the current imaging approach. In other words, the "discrepancies" found in this manuscript are primarily due to the resolution ability difference with those methods, which is fine and should be simply said it is so.

- *The method we are using is a modification of the S-receiver function method and has the same well known advantages and disadvantages   relative to other seismic methods. The advantages of this method as presented by Kind et al. (2020) rely on the fact that this approach does not lead to any acausality and is free from sidelobes prior to the signal. When we compared our results with the results of other methods, we possibly did not point out sufficiently the differences in resolution relative to P-receiver functions or controlled source seismics. Longer period S waves see more larger scale structures whereas shorter period P receiver functions see more local details. We will make this more clear.*

  We added accordingly some more explanations to the method.

- 2. The Introduction section is too succinct. While the paper title suggests results are on the greater Alpine area, the introduction

(and the discussion) is almost only on the Eastern Alps. Moreover, references to earlier results from S-to-P converted wave studies are completely missing, including those co-authored by the first author of this manuscript (e.g. Geissler et al. 2010). On line 36-37, the sentence "Most controlled source… of the European plate." is debated, as you also know; moreover, it depends on longitude which direction of subduction is predominant in the results. Line 44 should also list the EASI experiment.

- *We will skip the word "greater" in the title. We will also include in the discussion results obtained by Geissler et al. 2010 in the Alpine area. We will specify the sentence "Most controlled source..." and also mention the EASI experiment as you suggested.*

-

*We added the paper of Geissler et al. to the discussion and found that their results are in agreement with our results about about velocity reductions below the Moho in the eastern Alps. We thank the reviewer for pointing out this paper.*

3. The Data and Method section is way too succinct and does not provide all details that would allow reproducing these results or to apply them elsewhere. The number of used stations should be stated, together with the time frame of data collection. Same goes for the teleseismic earthquake catalogue, and whether a maximum magnitude was also used for event selection? Line 59 should give the exact time window for signal and for noise, as well as what is meant under "corrected for the sign of onset": is it absolute amplitude, or all polarities the same? Line 62 should mention over which time window was the P component amplitude measured, and how (maximum?). Why is this measured from -50 to -10 s when most of the signal is closer to the S arrival than -10 s? The subsequent sentence (L63-64) requires a reference. Line70 should specify for which phase and how the moveout correction is applied.

- *We think practically all parameters required for reproduction are given. All networks contributing data are mentioned in the Acknowledgement. These data are accessible via EIDA. All data until September 2020 have been used. The magnitude limit used for copying data is given. It is also stated that the criteria for selecting traces is the signal-to-noise ratio of SV, not the magnitude (note phase picking is only done on the individual SV component). We*

*applied the automatic phase picker within a time window of -50 to 20 sec of the theoretical SV arrival time. An onset was declared if the signal-to-noise ratio stayed for 3 sec above 6. If the maximum of that signal was negative its sign was changed. We also permitted only onsets closer than 10 sec to the theoretical onset. Lining up and summing traces along the picked onsets produced clearly superior results compared to lining up along theoretical travel times. This procedure was checked for many examples with satisfactory results. All SV amplitudes are normalized to 1 before summation, the P traces are normalized accordingly. Amplitude scales are given in the figures in % of the incident SV signal. We applied an additional selection criterion depending on the noise on the P component (L63-64). To our knowledge there exists no reference for such a criterion. We will in the manuscript extent these explanations. Moveout correction is a standard method in receiver functions and does not need an additional description.*

- *We think that all the really required informations have already been given in the first version.*

4. The authors aim at using as little as possible processing steps, inluding no filtering. This is very interesting, and calls for an explanation how different seismic sensors are treated? These are known to have different lower corner frequencies (120s, 60s, 30s, and also other values) -- can they all be used? If not, what sensors or frequencies qualify for this method to be applied on their data? Alternatively, is it in the nature of teleseismic S waves that one can stack these signals seemingly so easily? If so, what is their frequency content and are they similar for events at different distance and different magnitude -- can you please illustrate that with suitable figures? Finally, the disadvantages of no filtering and no deconvolution should be also stated.

-

*We used only broadband data with a flat response in the period band of our signals, which is about 3-5 seconds. Therefore, no correction is required. Since we are stacking many traces from very different sources, but recorded by relatively closely spaced receivers, source effects are largely averaged out (also effects of the near source structure). The signal form of the stacked signal is probably mainly determined by the average mantle attenuation and the average near receiver structure. Signal forms do not change*

*much in the distance range we used (SVdiff is not used). Large magnitude events with complicated waveforms are relatively rare. There is a figure in Kind et al. 2020 showing examples of waveforms before stacking. Deconvolution was originally introduced to equalise waveforms before stacking. This improves the signal-to-noise ratio especially in shorter period P receiver functions, not that much in longer period S receiver functions. Deconvolution is a none-unique approximate method with several parameters to choose. It is not free of side effects. Also deconvolution needs a relatively large time window and declares everything within this window as source signal. This, for example destroys small precursors of larger signals which might carry important information.*

- *Plain wave stacking (without deconvolution) is done since a long time (Shearer 1991) with long period teleseismic data (Bodin, Yuan and Romanowicz 2014, or more recently by Kind et al. 2020 and references therein, or Liu and Shearer 2021).*

- *Our paper is not meant to be a methodological paper. It has rather the intention to contribute to solve geological questions. Therefore, we refer to earlier papers for modifications of the S receiver function method.*

- *We added some references about stacking seismograms without deconvolution.*

- 5. One source of possible artifact of waveform amplitudes is the rotation to the LQT coordinate system, which depends on the ray geometry and therefore the velocity at the surface. Applying the iasp91 model is therefore a simplification, as there are (locally deep) sedimentary basins throughout the study area. Please estimate the uncertainty due to the use of iasp91 instead of local velocity values in this processing step.

- *Theoretical rotation angles of a 1D model are regularly used in many receiver function papers dealing with deeper structures. Exceptions are anisotropy studies or specific small scale near station heterogeneities with good azimuthal coverage. Sedimentary basins would change mainly the incidence angle, not much the backazimuth. Most S receiver function studies use anyway the vertical component and not the L component. According to our experience using theoretical rotation angles does not change much regarding the stacked Moho or upper mantle signals.*

- *Our method is identical with the well known S-receiver function method except that any alterations of the waveforms are omitted and that as reference times for stacking the SV onsets are used and not the maxima of the deconvolved SV signals. Rotation, moveout correction and migration are identical and have been discussed since a very long time.*

6. Onset time selection. Line92 says that "the arrival times of the seismic signals must be picked … at the beginning of the signal…". However, this onset time is frequency dependent, and stacking waveforms with different frequency content (either due to sensor characteristics or/and to earthquake spectra) is therefore error-prone. Here the authors chose not to apply any filtering, yet it remains to be shown that individual waveforms contributing to these stacks have similar dominant frequencies (see also comment 4 above), so that a "stack-onset" makes sense. See also point 8 below on noise levels.

- *We are picking SV arrival times on individual traces (not on summation traces) of large signals with an established method, which is used at many places. So far we did not pick Moho arrival times (on the P component) of summed traces (with the exception of the estimates of the largest depths in Fig.18). This is indeed problematic in some regions because of weak Moho precursors, which are possibly related to additional velocity increases below the main Moho discontinuity. Therefore, we are working now on comparing SV waveforms with Moho waveforms for the determination of the correct arrival time. The dashed lines in the profiles are not meant to mark arrival times, they mark Moho depth changes along the profiles. For stacking of plain broadband waveforms see response to comment 4).*

-

- *We added more references about wave form stacking in the revised manuscript.*

7. The interpretation of depths can only be done very approximately, for two reasons: (a) the corresponding waveform stack is done at a constant depth of 50 km across the whole region for the

Moho, and (b) the time-to-depth conversion uses a 1D global velocity model (iasp91). The results themselves showing Moho depths ranging from ca. 20 to ca. 65 km, one can easily see the deviations that apply horizontally (for a) and respectively vertically (for b). This limitation should be spelled out in the manuscript, possibly at the same place where the method's applicability is discussed (see comment 1 above).

Even more important would be to estimate the depth uncertainty of this method, primarily by the use of a 1D velocity model for time-to-depth conversion, and also due to the uncertainty in picking onsets on the waveform stacks + incoherency between neighbouring stacks simplified in the dotted lines added by hand over the general trend. Depth differences compared to previous results that are less than this uncertainty should not be reported as surprising of different.

- *Yes, we stacked traces with approximate S-to-P piercing points at 50 km depth (using the iasp91 model). Changing that piercing point depth by a few tens of kilometres does not affect the results significantly. Depending on the distribution of stations, noticeable differences could arise by comparing stacks with about 100 km difference in piercing point depth. Most papers with time domain stacking of Moho signals are not using piercing point depths. This is ok for single station, but it is not completely correct for closely spaced stations. For example Hetényi et al. (2018) did not stack traces by piercing points but by stations, although traces recorded at neighbouring stations are overlapping. This means zero km piercing point depth and not a piercing point depth near the Moho depth. The effect is noticeable but not very large. Doing this leads to some kind of smearing. We decided to use piercing point depths also for the Moho to be more correct.*

- *The problem of using a 1D model for depth estimates is clear. We leave using 3D model for a later time. We will point out also more clearly, that lateral arrival time variations may also be explained by changes in average crustal velocity.*

-

We have pointed out more clearly the possible influence of lateral velocity changes on depth estimates in the new version.

8. Negative velocity gradients (NVGs) are mentioned on Line 97 as

red signals, and these are present on most profiles below the Moho peak. What is not clear is the extent to which these red signals are interpretable: some of them are clear and high amplitude, some others are poorly defined or even opposite polarity. What is the threshold limit, the noise level, above which an NVG (or Moho) can be interpreted? Line 170-171 says amplitudes of ~10% and <4% are clear for Moho resp. NVG, where is the noise? At 1%? Can the onsets on the qualifying traces be highlighted with a dot to better support the interpretation of NVGs?

On many occasions the onsets are picked for a wave of tiny amplitude… Also, Line 97 says these NVGs are sharp or gradual velocity change -- could you please quantify? How sharp they can be with respect to the waveforms we see for the Moho? How gradual can they be considering the waves' frequency content?

*There is a clear difference between the Moho signals and the NVG signals. The Moho signals mark a laterally continuous discontinuity and NVG signals form clusters of signals. It is difficult to decide which data are significant for the interpretation and which ones are not. We decided to discuss some of the few most obvious clusters. We think marking NVG onset times in some traces does not make very much sense since there is very much scattering. All that can be said is that in these clusters downward velocity reductions occur. Li et al. 2007 have discussed the sharpness a possible LAB observed with S receiver functions. They found that it is difficult to differentiate between zero and about 20 km thickness of the discontinuity.*

We added the reference Li et. al.

INTERPRETATION CONCERNS WORTH RECONSIDERING OR REVISING

-Line 113-115: the Adriatic Moho "rise" on profiles 2 to 6 is overinterpreted and should be removed based on the following justification. On profile 2, not all onsets are followed by the dotted interpretation line. On profile 3, there are 0.6° without data, and the next stack (at 45.2°N) does not show the Moho onset where it is picked. Profile 4 has an even larger gap and no data where the "rise" is drawn. Profile 5 also suffers from the data gap and the dotted line does not follow the onsets closely, the Adria Moho onsets reach as deep as the European one! Profile 6 has more than 1.0° data gap and therefore the interpretation of the rise is not supported.

- *We will describe this in more detail according to your comments*

- We made according corrections.

- -Line125: results here show 65 km depth, though Spada reaches clearly less! Please elaborate on the depth uncertainties of your method, as proposed under comment 7.

- *This is probably a question of the waveform of the Moho signal. Precursors from a weaker discontinuity could be the cause. For this reason we will do a waveform comparison between Sv and Moho signal.*

- We discussed the question of Moho precursors in much more detail in the revised manuscript.

  -Line129-133: these observations make sense, and they seem to concur with Bruckl et al.'s results on the Pannonian fragment starting already in the Eastern Alps; it is worth citing this here.

- *We will do this*

- We have cited Brückl et al. here and also pointed also to a possible relation of their boundary between the newly defined Pannonian lithosphere and the European lithosphere with our to the northward dislocated Moho trough east of about 13°E.

  -Line139-145 on the MHZ: the "jump" and the "suddenness" are not justified by the data presented here and should be taken with a pinch of salt or two, see the following arguments. First, the presented profile is 2.0° wide, along which the MHZ changes its latitude significantly (~0.5°), and so quite a bit of lateral variations are projected (smeared) onto the profile. Second, 2 bins are lacking from the profile, making a gap of at least 0.4°, so no sharp change can be resolved in this area. Third, on Figure 11A, the MHZ is shown to separate the "Eu Moho" from the "Pa Moho" while it separates ALCAPA and Tisza-Dacia blocks. Fourth, your own comment on velocities (Lines 143-144). Fifth, all previous investigations, from the earlier active seismic results (oil industry and research profiles such as CELEBRATION and those compiled in Horvath et al.) to more recent local RF studies and ANT studies (e.g. Szanyi et al. 2021) show there is no significant jump of Moho depth across the MHZ. The approach presented here is resolving much coarser structures and therefore the interpretation of a large offset across a sharp fault does not stand.

- *You convinced ourselves that the discussion of the resolution of the Moho at the MHZ in comparison with earlier results should be improved. We will try to go into more details, also on the expenses of lower signal-to-noise ratio.*

-

  This part is completely changed. In the Supplement we added some narrower profiles, which have, however, a lower signal-to-noise ratio. We mention another interpretation of the apparently deeper Moho north of the MHZ. A second discontinuity below the Moho could be responsible for the early arrivals. But this interpretation needs to be checked by a waveform comparison, which will be done in a later study.

  -NVG interpretations. As alluded to above, the NVG onsets are less clear and the interpretations seem to be very subjective. (This is also reflected in the author's own statement in the Supplementary Material text, paragraph 2, last sentence -- why are these signals less correlated?)

  For example, on profile 1, the northernmost 4 stacks do not have an onset where it is interpreted to have one with the gray lines; stacks number 7, 9 and 11 from north (left) have blue signals instead of red. So for profile 1, the NVG is not supported or at least not clear at all from the data. Profile 2 has numerous stacks where the gray lines are not over the red wave onset (around 47°N). Profile 3 also has a few stacks that are incoherent with the gray lines. On profile 4 the gray line interprets 10 stacks, out of which at least 4 have no clear red wave onset. Profile 6 also includes stacks not fitting the interpreted picture. Line 186 says there is no significant NVG in profile 8, but one could draw it, from -7s at 50°N to -13s at 48°N! Profile 11 shows an E-dipping NVG, but a nearly symmetrical one can be drawn, from -12s at 12°E to -7s at 18°E! Profile 12 has no NVG drawn, but one could also draw one there...

- *We now try to make the aspect of the drawing more homogeneous.* I hope these examples demonstrate the subjectivity of interpreting NVGs. Maybe it is more constructive to think of alternatives? For example, my visual impression is that sub-Moho negative amplitudes are higher on the European plate than on the Adriatic plate. Could this be checked and quantified? For example, above a set noise level (e.g. 1% amplitude?), one could show the max.amplitude of sub-Moho local minima on a map? Maybe even further details will show up.

- *The NVGs are difficult to quantify. We marked the bottom of the NVG region by grey scattered lines. Picking first arrivals of the NVGs is more difficult than picking Moho arrivals. We are working in a subsequent manuscript on comparing Moho and NVG waveforms with SV waveforms for more reliable determination of Moho and NVG arrival times. With such improved data we will make a Moho map and try to make a NVG map. Especially for the NVGs it is not sure if significant improvements can be achieved. In the present paper we can only point out the existence of such zones with hints of their possible meaning.*

-

We adjusted the grey lines in several profiles. We plan further discussion in an upcoming paper.

-Under "Comparison with earlier results" the new results are compared with Heteyni et al 2018 (abbreviated here as H2018) and the comparison could be a bit more complete. Line222 says that the new results do not show that the Adriatic Moho reaches 70 km depth. The H2018 results argue for a broad vertical gradient zone, from 50 to 70 km depth, and use proper migration with multiples, while the new results do not; this could be mentioned. The general agreement (Line 223) is actually encouraging. Line 226-227 says "Our data do not support the postulate of Hetényi et al. (2018b)

that the Adriatic Moho in the Eastern Alps dips northward underneath the European Moho." The explanation is relatively easy, as H2018 used a narrower profile width, while the new results here cover 1.0° width, from 12.5 to 13.5°E; and since signficant lateral variations are now known in this part of the Alps, it is therefore not suprising that you find a symmetric shape, as in Spada et al. results, simply because many rays sampling farther west are included.

- *Our results indeed agree nicely with H2018 with the exception that we do not see the plunging of the Adriatic Moho beneath the European Moho. This may be a question of our lower lateral resolution. However, the H2018 data also do not appear to be extremely clear.*

-
    We pointed out that H2018 use higher resolution P-receiver functions.

-
    Our method is identical with the well known S-receiver function method except that any alterations of the waveforms are omitted and that as reference times for stacking the SV onsets are used and not the maxima of the deconvolved SV signals. Rotation, moveout correction and migration are identical and have been discussed since a very long time.

-

- -still under Comparison, the Paffrath et al. results are shown. First, these are still in review, and not yet final. Second, it is extremely hard to see their results on your Figure 17, the contour label texts of the tomography are very small and sometimes broken over two lines. Please improve this figure. Finally, the tomograpy results show a major gap of the high-velocity anomaly at 8.5°E. This should be mentioned and discussed in the interpretation, see e.g. Line238. Line247 refers to the Tauern Window, please report its location above the profile on Figure 17B.

- *Thank you for pointing out that Fig. 17B is indeed very difficult to read. We improved this figure for clarity by increasing the size of the numbers denoting p-wave velocities*

- *Figure was changed*

-

- -at the comparison with Bruckl et al. 2010 profile at 13°E, the original publication shows line drawings for plates, but also the location where original Moho depth data is taken from the map of Behm et al. 2007. It would be better to cite the original data, and show where that data shows Moho. The gap present in that dataset would allow a bit broader range of interpretations. In the same paragraph, comparison to Kummerow et al. is made, and qualified "fair". Locally >5 km depth difference is present with their results, although this is not surprising, as TRANSALP was located farther to the West then this profile, please mention this. In the same paragraph, comparison with Spada et al. 2013 is made (Figure 18B). The interpretation of this profile without Spada et al.'s results overlaid would actually be more difficult. Moreover, Spada et al.'s Moho depth is the same shape but clearly shallower than the onset of blue phases. Is it an indication of noise level, to be distinguished from onset times? Or some other effect?

- *We will make these changes. Unfortunately it is not clear what causes the difference with Spada's  model in the north. Perhaps a difference in the crustal model together with the difference in the methods.*

-
  We added the reference Bruckl et al. and Behm et al. to this discussion and pointed out the different longitude of the Transalp profile.

  -figure 20B, comparison with Grad et al. 2009a: here again, the Moho by Grad et al. is systematically shallower then the newly presented results. Why? (See detailed question above). And why is the eastward deepening more pronouned (L289)?

- *Again, the reason for the difference between the Grad Moho and our results is not clear. It could again be differences in the crustal model, the method used. We also think we need to check our results with a waveform comparison of the SV and Moho signals. The difference of the two east-west profiles in Fig.20 is that the southerly one is in the Pannonian Basin and the northerly hits the western Carpathians.*

- We plan to work on this problem in a subsequent paper. We need to study in more detail the influence of Moho precursors on the determination of Moho onset times.

SOME MINOR COMMENTS

- *We will respond carefully to all the following comments and change most formulations as suggested by the reviewer.*

-Line 17: insert "depth" after "Moho"   This is done

-L19: "mantle trench" does not really make sense, if anything it is a crustal trench but even that reads odd   This is changed

-L19-20: "where the Eurasian lithosphere is subducted below the Adriatic lithosphere": please specify the region where this is meant -- it seems like this is in the Western to Central Alps? (In the Eastern Alps it is contested, as you also say it, so without geographical specification this sentence is bizarre in the abstract)

- This is done done

-L21: "updoming" suggests there is dynamics, a movement of the Moho, while the structure you image is static. Can you use another word? See also Line302 and 329. This was changed

-L22: "into" → "to". By the way, this shallower Moho is already part of the Pannonian fragment according to Bruckl et al. See also Line 329.
We referred to the Pannonian fragment of Brückl et al.

-L23-24: "negative P-wave velocity gradient": indicate the source of this information
this is one of our results

-L26: please replace commas with dashes in "Alpine, Carpathian, Pannonian"   done

-L121: Yuan et al. 1997 actually says "The Indian lower lithosphere is considered to underthrust the Asian crust to the Banggong suture [Ni and Barazangi, 1983; Beghoul et al., 1993; Owens and Zandt, 1997], which is beyond the present INDEPTH transect." so it may be more appropriate to cite one if these references? By the way, reading lower lithosphere here, and checking the Nabelek paper as well, "Indian crust" on line 121 would be better as "Indian lower crust and mantle" or "Indian lower lithosphere". *Will be done.*
We added the Nabelek reference

-L135 and section on the "new" Moho depression. It is the main result of the manuscript, and a comparison with earlier Moho maps, e.g. from Grad et al., or earlier S-to-P RF results, would be worth adding to the discussion.

-

We compared our results with those of Grad along a profile. So far we have no Moho map, but we are working on it.

-L148: please mark this longitude on top of the figure as an important boundary with Eu and Ad labels on either side.

- This point is not clear to us. The boundaries we used are marked on the Spada map.

-L149: does the eastern end of this profile reach the Pannonian? From the other (N-S) profiles it seems so; please mark that boundary as well.

- yes, it would reach the new Pannonian part.

-L151: less than 20 km depth seems extreme, no previous study reported such thin crust in the Pannonian. Maybe mention depth uncertainties here due to the applied velocity model with respect to reality.   done

-L160: please mark this longitude on top of the figure as an important boundary with Eu and Ad labels on either side.

- It is not clear to us what you mean with this remark.

-L176: "onset" instead of "arrival" seems to be a better choice following your earlier argument. Please append "in this time window" at the end of the sentence.   done

-L193: citing some of the MT results for thin lithosphere in the Pannonian Basin would strengthen this statement.

- We preferred not to start this kind of discussion at this at this time

-L194: remove reference to 10 km step across the MHZ (see interpretation comment above).   done

-L199-200: because of the gray overlay one cannot see the wave-forms to check the interpretation of the NVG… see also general reservation on NVG above.

- The grey lines mark the approximate start of the extended red wave groups, not really onset times

-L203: see comment on possibility to draw symmetric NVG on profile 11.

- Our point is that the marked one is much stronger than the possible symmetric one

-L204: see comment on possibility to draw NVGs on profile 12.

- We think there is too much scatter in this profile

-L205: how steep is this "relatively steep dipping structure", considering that the profiles are ca. 9-fold exaggerated vertially? My estimate: about or less than 10°.

- yes, but the uncertainty is high and the dip angle may be biased in receiver functions

-L211: it may be worth adding that it is in the vicinity of the contact with Adria, but not crossing it!

- it is parallel to the contact with the Adria plate

-L231: replace "data, and those of" by "data, similarly to those of" (to clarify the sentence)  done

-L245: the word "rise" (two occurrences) seems to suggest there is dynamic movement, is there a better expression?  changed

-L267: Hetenyi et al 2015 have no SW-NE profile, their three profiles are NW-SE. Please correct.   done

-L270 and 275: Handy et al reference year should be 2021? changed

-L283: "although there are some differences in details" is very gentle, as locally there are >10km differences (lon.9°E, jump at 14°E). What is the point showing this comparison of the differences are not discussed?
This is an early and basically correct refraction result which should be mentioned

-L299-300: important along-strike changes in Moho topography were also highlighted in the EASI results (H2018), worth citing?

•

EASI is NS striking??

-L304: shallower Moho, continuing to the Pannonian, already proposed by Bruckl et al., worth citing?  done

-L304-305: this result is contested and should be removed, or reworked to one that is supported by data and data coverage.

• done

-L306: "in front of" would be better as "north of"?   changed

-L315: "rapid jump" seems to refer to a motion, but the image is

static. See also L 334-335. "Sudden depth change"? Also, would it be worth comparing this result with one of the CELEBRATION lines?

line 315 is changed, Hrubcova is now cited

-L323-324: "help to increase the uniqueness" -- in this study there is no demonstration of uniqueness… it could be done (by boot-strapping?)   changed

-L332: Mroczek et al. seem to claim the same, worth citing?

•

Has been cited in Fig. 17

-L336: mantle → asthenosphere?   done

-Conclusions Line339-347: as written above: point 1 is not supported by the data and should be removed, point 2 is a good result, point 3 needs to be strenghtened.
Point 1 is completely changed. In point 3 a reference to the according figures is given.

-L399: "Alparray" → "AlpArray Seismic Network"   changed

-Figure 7A profile 5: the "Li Moho" is interpreted at an unusually short wavelength over 4 traces, different from other interpreted lines. By simply connecting onsets a much flatter line should be drawn.   changed

-Figure 9A profile 7: the black arrow is very hard to see. The "culmination" in the caption is already the Pannonian fragment discussed by Bruckl, also Mroczek. The "Moho gap" in in Spada's

work, H2018's work is not at this longitude and should be removed.
black arrow is changed to white

-Figure 10A profile 8: the black arrow is very hard to see. Some of this profile (the shallowest part) could be Pannonian Moho, if Bruckl et al's interpretation is followed.
Arrow is white now. Brückl's definition of a Pannonian lithosphere is cited.

-Figure 15: the overall depth uncertainty should be quantitatively mentioned in the caption.   done

-Figure 16 the cyan quadrant is missing from the map.   corrected

-Figure 19B: the reference below this panel is not Schmid et al. 2020 but Handy et al. 2021, most likely   changed

We wish to thank both reviewers for their really helpful comments.

---

## Author Response (AR2)

Response to the Reviewer

We responded to the requests accordingly